# Black-Box Forgetting

**Yusuke Kuwana**
Tokyo University of Science
4624513@ed.tus.ac.jp

**Yuta Goto**
Tokyo University of Science
4623511@ed.tus.ac.jp

**Takashi Shibata**
NEC Corporation
t.shibata@ieee.org

**Go Irie**
Tokyo University of Science
goirie@ieee.org

## Abstract

Large-scale pre-trained models (PTMs) provide remarkable zero-shot classification capability covering a wide variety of object classes. However, practical applications do not always require the classification of all kinds of objects, and leaving the model capable of recognizing unnecessary classes not only degrades overall accuracy but also leads to operational disadvantages. To mitigate this issue, we explore the selective forgetting problem for PTMs, where the task is to make the model unable to recognize only the specified classes while maintaining accuracy for the rest. All the existing methods assume "white-box" settings, where model information such as architectures, parameters, and gradients is available for training. However, PTMs are often "black-box," where information on such models is unavailable for commercial reasons or social responsibilities. In this paper, we address a novel problem of selective forgetting for black-box models, named Black-Box Forgetting, and propose an approach to the problem. Given that information on the model is unavailable, we optimize the input prompt to decrease the accuracy of specified classes through derivative-free optimization. To avoid difficult high-dimensional optimization while ensuring high forgetting performance, we propose Latent Context Sharing, which introduces common low-dimensional latent components among multiple tokens for the prompt. Experiments on four standard benchmark datasets demonstrate the superiority of our method with reasonable baselines. The code is available at https://github.com/yusukekwn/Black-Box-Forgetting.

## 1   Introduction

Large-scale pre-trained models (PTMs) such as CLIP [Radford et al., 2021] and ALIGN [Jia et al., 2021] have strong capabilities of zero-shot classification for everyday objects. Nevertheless, in practical applications, the classification of all kinds of object classes is rarely required. For example, in an autonomous driving system, it would be sufficient to recognize limited classes of objects such as cars, pedestrians, and traffic signs. We would not need to recognize food, furniture, or animal species. Retaining the classes that do not need to be recognized may decrease overall classification accuracy, as well as cause operational disadvantages such as the waste of computational resources and the risk of information leakage. In this paper, we address the problem of selective forgetting of specified classes [Shibata et al., 2021, Graves et al., 2021, Ye et al., 2022, Tarun et al., 2023], i.e., tuning a pre-trained model to reduce the classification accuracy for only the specified classes with-

38th Conference on Neural Information Processing Systems (NeurIPS 2024).

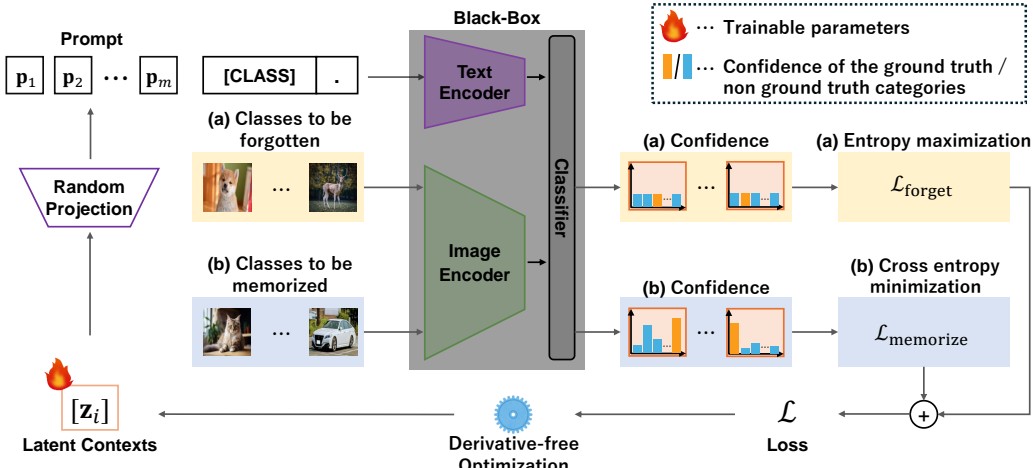

**Figure 1: Overview of our black-box forgetting framework**. The confidence of each class is computed as the similarity with the image and class (text) embeddings from the black-box pre-trained vision-language model (e.g., CLIP). The obtained confidence is used to compute the respective loss functions for the classes to be forgotten and the classes to be memorized. (a) For the classes to be forgotten, maximize the entropy of the confidence so that the accuracy is reduced. (b) For the classes to be memorized, minimize the cross-entropy loss to retain the accuracy. These two objective are jointly optimized to tune the learnable text prompt. The gradients of the objective are not available when the model is black-box. We therefore use CMA-ES [Hansen et al., 2003], a derivative-free optimizer, to learn the text prompt. Instead of directly optimizing the original high-dimensional context (token) embeddings for the prompt, our method learns lower-dimensional latent contexts for mitigating the difficulty of high-dimensional optimization.

out affecting the accuracy for the others[1]. While selective forgetting of specified classes has long been overlooked [Ye et al., 2022], a few existing methods have been proposed very recently [Shibata et al., 2021, Ye et al., 2022, Tarun et al., 2023]. The seminal work is Learning with Selective Forgetting (LSF) [Shibata et al., 2021], which has been proposed in the context of continual learning [Kirkpatrick et al., 2017, Li and Hoiem, 2017, Aljundi et al., 2018] and uses a special random code called mnemonic code to control the class-wise memorization and forgetting. A similar idea has been proposed to achieve forgetting by learning noise that maximizes the classification error for the classes to be forgotten [Tarun et al., 2023]. An extended version of LSF [Ye et al., 2022] allows forgetting of specified classes as well as recovery of them by temporarily transferring the knowledge of the classes to be forgotten to another network called deposit module.

Overall, all the existing methods assume the "white-box" setting, where the complete information of the target model is available for training/tuning, including the model architecture, its parameters, and their gradients. However, major PTMs such as GPT-4V [OpenAI, 2023] are often "black-box," where the model itself or its information is often fully or partially private due to commercial reasons or considerations of social impact. Since the parameters and their gradients are not accessible in such a model, all the existing methods are inapplicable. To the best of our knowledge, selective forgetting methods for black-box models have never been studied to date.

In this paper, we address Black-Box Forgetting, i.e., the selective forgetting problem for black-box PTMs, and propose a novel approach to the problem. Given the unavailability of model information, our method, unlike the existing selective forgetting methods, does not optimize network parameters nor utilize the gradients of the parameters; we instead optimize the input textual prompt to decrease the classification accuracy of specified classes to be forgotten in a derivative-free optimization framework. One disadvantage of derivative-free optimization would be that it is not effective nor efficient for high-dimensional problems due to the low convergence rate in high-dimensional spaces [Qian et al., 2016], and unfortunately, the textual prompt is typically parameterized as a set

---

[1]Note that the problem focused on in this paper is closely related to but different from the typical "machine unlearning" problem, which is the task of removing an arbitrary sample from a pre-trained model, i.e., obtaining a model that is identical to the model trained from scratch without that sample [Cao and Yang, 2015, Golatkar et al., 2020a, Sekhari et al., 2021, Bourtoule et al., 2021, Golatkar et al., 2021, Kurmanji et al., 2023].

of high-dimensional vectors in PTMs, e.g., 512-D for each "context" (i.e., learnable token in the prompt) in CLIP ViT-B/16 [Dosovitskiy et al., 2021]. To mitigate this issue, we propose Latent Context Sharing (LCS), a novel parametrization method of the contexts. The core of LCS is to parameterize each context with low-dimensional latent components, which consist of token-specific components and common components among multiple tokens for the prompt. Experimental results on four standard benchmark datasets demonstrate that our method improves zero-shot CLIP and outperforms reasonable baselines based on black-box prompt tuning [Sun et al., 2022b].

The main contributions of this paper are summarized as follows:

- We introduce Black-Box Forgetting, a novel problem of selective forgetting for black-box models.
- We propose a novel method for Black-Box Forgetting based on derivative-free optimization of learnable text prompt.
- We introduce Latent Context Sharing (LCS), a novel parametrization method of contexts for mitigating the difficulty of high-dimensional optimization with derivative-free optimization.

## 2 Related Work

**Machine Unlearning**   Machine unlearning aims to remove an arbitrary sample from a pre-trained model, i.e., obtaining a model that is identical to the one trained from scratch without that sample [Cao and Yang, 2015, Golatkar et al., 2021, Sekhari et al., 2021, Bourtoule et al., 2021, Kurmanji et al., 2023, Guo et al., 2020, Chen et al., 2019]. Many methods have been proposed, for example, to construct a forgettable model by transforming the learning algorithm into a sum of the training samples [Cao and Yang, 2015], to achieve forgetting by linear approximation of a nonlinear model [Golatkar et al., 2021], and to update the model to be closer to / farther from the original model in the retain / forget samples [Kurmanji et al., 2023]. Methods specific to certain learning algorithms such as LDA [Guo et al., 2020] and SVM [Chen et al., 2019] have also been explored. Machine unlearning and Black-Box Forgetting are closely related but different; Machine unlearning aims to remove the influence of specified training samples on the training model, whereas Black-Box Forgetting aims to prevent the recognition of specified classes. Forgetting specified classes has attracted much attention recently in various contexts [Heng and Soh, 2023, Lu et al., 2024, Zhang et al., 2024, Shibata et al., 2021, Ye et al., 2022]. We in this paper address the black-box setting, which has not yet been explored.

**Selective Forgetting.**   Shibata et al. [2021] proposed Learning with Selective Forgetting (LSF), which updates the model for a new task by forgetting only certain classes from the previous task while memorizing the rest of the classes. Golatkar et al. [2020a] introduced a scrubbing method that involves a shift in weight space and addition of noise to the weights to remove information from network weights. They also proposed a forgetting mechanism to linearly approximate the weights that would have been obtained by unlearning [Golatkar et al., 2020b, 2021]. Tarun et al. [2023] proposed an error-maximization-based method to learn a noise matrix for the class to forget, and the model is updated by training on this noise. Then, fine-tuning on the classes to be memorized to adjust the model weights.

Since these methods require the model weights or the gradient of the model parameters, they cannot be applied to black-box models. In this paper, we introduce a new selective forgetting method for black-box models that does not require the model weights or the gradient of the model parameters.

**Black-Box Learning.**   Black-Box Tuning (BBT) [Sun et al., 2022b] is a black-box prompt tuning method for large language models. BBTv2 [Sun et al., 2022a] improves BBT with deep prompt tuning. They achieve accuracy comparable to white-box learning methods in various natural language tasks. BDPL [Diao et al., 2023] fine-tunes a collection of discrete prompts for language models by treating the word choice in the prompt as a reinforcement learning policy.

BlackVIP [Oh et al., 2023], the first black-box learning method for vision-language models, optimizes a generative model that generates visual prompts embedded in images by zeroth-order optimization. LFA [Ouali et al., 2023] extends the capabilities of black-box models by assuming access

to pre-computed features from pre-trained backbones. Through a multi-stage procedure, it optimizes a projection layer to enhance alignment between pre-computed image features and class prototypes. Guo et al. [2023] introduced a collaborative black-box tuning (CBBT) for optimizing both the textual prompt and adapting output visual features in black-box vision-language models. The textual prompt is optimized by estimated gradients and the visual adapter is trained through direct supervised learning from the output features.

In this study, we focus on textual prompt tuning for the black-box model and introduce Latent Context Sharing (LCS), which improves accuracy while reducing the number of dimensions to be optimized.

# 3 Method

The overview of the proposed method is illustrated in Fig. 1. We use the vision-language model CLIP [Radford et al., 2021] as the base model and optimize an input prompt for the CLIP text encoder based on the loss that requests reduced accuracy of selected classes. The derivative-free optimization method must be used to optimize a textual prompt for the black-box model where the gradients of its parameters are unavailable. We employ CMA-ES, a widely used evolutionary algorithm for black-box optimization in continuous, because a textual prompt to be optimized is a continuous variable. CMA-ES is a multi-point search algorithm based on a multivariate normal distribution and proceeds the search by iterating (i) sampling candidate solutions, (ii) evaluating the loss values of the candidates, (iii) weighting the candidates based on the loss values, and (iv) updating the mean and covariance matrix of the distribution by using the weighted candidates. Due to the nature of multi-point search, the performance of CMA-ES degrades in high-dimensional problems, typically ten or more dimensions [Ros and Hansen, 2008, Akimoto and Hansen, 2016]. While several extensions have been proposed, e.g., [Ros and Hansen, 2008, Akimoto and Hansen, 2016], these methods require knowledge of independence among variables, which is not always known. In this paper, we propose a customized extension of CMA-ES to Black-Box Forgetting. In general, when applied to high-dimensional black-box continuous optimization by CMA-ES, the computational complexity can become a hindrance. The key is reducing the dimensions of the latent variables in the textual prompt while preserving their context representations.

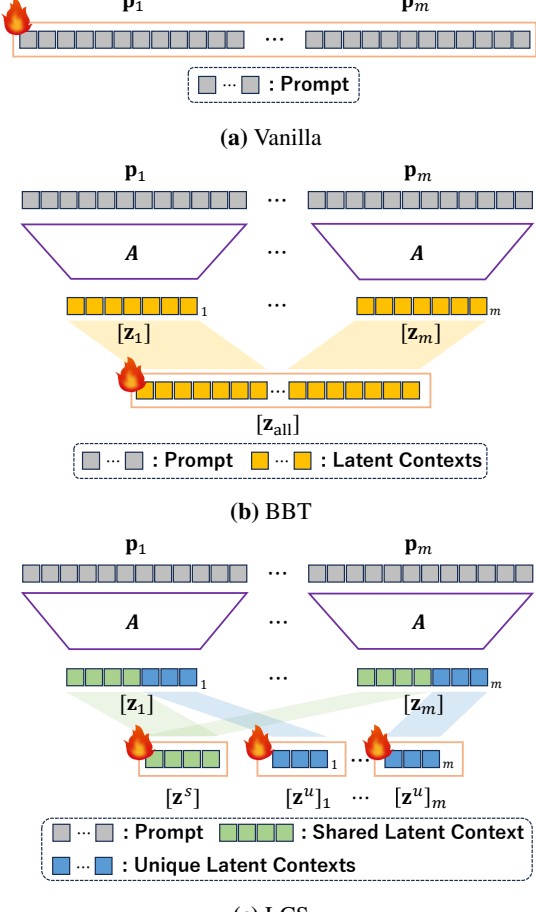

**(a)** Vanilla

**(b)** BBT

**(c)** LCS

Figure 2: **Comparison of context parametrization.** (a) Vanilla prompt tuning optimizes the textual prompt directly. This approach requires high-dimensional optimization. (b) BBT [Sun et al., 2022b] optimizes a lower-dimensional latent context instead of directly optimizing textual prompt to mitigate high dimensionality. (c) In our LCS, for more effective optimization, a latent context is composed of unique components and common components among multiple latent contexts, and each component is optimized independently.

### 3.1 Context Parametrization

We discuss two types of context parametrizations: i) Latent Representation with Random Projection for Black-Box Tuning (BBT) [Sun et al., 2022b] as preliminary; ii) Latent Context Sharing (LCS) for our method, a more effective context parametrization approach for the black-box forgetting.

**i) Preliminary: Latent Representation with Random Projection.** The dimension $D$ of a context in the prompt is extremely large, which makes derivative-free optimization difficult. To mitigate this high dimensionality, BBT introduces a low-dimensional latent context $[\mathbf{z}_{\text{all}}] \in \mathbb{R}^{d \times m}$, where $d$ is the dimension of a latent context and $m$ is the number of latent contexts. Then, BBT divides $[\mathbf{z}_{\text{all}}]$ into $[\mathbf{z}_i] \in \mathbb{R}^d$ and generates contexts for the prompt by projecting them to the original context dimension by a random projection $\mathbf{A} \in \mathbb{R}^{D \times d}$ (see Fig. 2b) sampled from a normal distribution $\mathcal{N}(0, \sigma)$, where $\sigma$ is the standard deviation of the context (token) embeddings. The dimension of variables to be optimized is suppressed more than optimizing a context directly because $d$ is a lower dimension than the original context dimension ($d \ll D$).

**ii) Latent Context Sharing.** As empirically shown later in Sec. 4.2, the effectiveness of the context parametrization for BBT described above is limited for selective forgetting settings. We propose latent context sharing (LCS), a more efficient context parametrization.

Fig. 2c shows the overview of LCS. The key idea is to assume shared parameters among different latent contexts. This inspiration comes from successful word embedding methods; most word embedding methods are trained on the assumption that locally co-occurring words have semantic correlations between them (e.g., [Milkolov et al., 2013, Pennington et al., 2014, Devlin et al., 2019]). This inspires the idea of explicitly modeling semantic correlations between words in a prompt as shared components. We assume that each latent context is composed of unique components (*Unique Latent Contexts* (ULC)) and common components (*Shared Latent Context* (SLC)) among multiple latent contexts. Then, we optimize each ULC and SLC independently. Each latent context $[\mathbf{z}_i]$ is obtained by concatenating SLC $[\mathbf{z}^s] \in \mathbb{R}^{d^s}$ and ULC $[\mathbf{z}^u]_i \in \mathbb{R}^{d^u} (i = 1, \cdots, m)$, where $m$ is the number of latent contexts, $d^s$ and $d^u$ are the dimension of SLC and ULC, respectively. Despite the number of parameters prepared for BBT and LCS is the same ($m \times d = d^s + m \times d^u$), LCS is possible to significantly reduce the number of optimization dimensions compared to BBT[2], because LCS optimizes each ULC and SLC independently. Compared to assuming that each latent context is completely independent (i.e., using only ULC), providing common components has the substantial advantage of not losing dependencies among multiple tokens for the prompt.

Note that, CoCoOp [Zhou et al., 2022a] also introduces an approach that incorporates a shared component in the context of the prompt to improve the generalization in the white-box setting. While CoCoOp learns the network that generates a shared component based on image features, our method directly learns the shared component. The optimization for our black-box forgetting using our method becomes simpler because the number of dimensions for optimization is minimal. As shown in the experimental results in Sec. 4, introducing a shared component is effective, which suggests that optimization of shared components has a impact in our problem settings.

### 3.2 Loss Functions

We apply different loss functions to the classes to be memorized and the classes to be forgotten. The cross-entropy loss is used for the classes to be memorized to maintain the classification accuracy:

$$\mathcal{L}_{\text{memorize}}(\mathbf{p}, \mathbf{t}, C) = -\sum_{i=0}^{C-1} \mathrm{t}_i \log \mathrm{p}_i, \tag{1}$$

where $C$ is the total number of classes, $\mathbf{p}$ is confidence of each class obtained by applying the Softmax function to the similarity of each class, which is the output of CLIP, and $\mathbf{t}$ is the one-hot vector of the label of an input image.

A naive approach to ensuring that selected target classes are forgotten is to reduce the confidence of that class in the input images of the class. In general, however, this naive approach leads to undesirable behavior in the model. For example, if we force the model to forget the class "dog,"

---

[2]The actual amount of calculation is $O(d^2)$ as CMA-ES estimates the covariance matrix internally.

it may lead to the model always classifying images of "dog" as the class with features similar to "dog," e.g., "cat." Moreover, such an unintentional bias may provide sufficient clues to identify the forgotten classes, and consequently may lead to the risk of information leakage. We, therefore, want to make the classification results for images with the forgotten classes close to random and exclude information about the classes. To this end, we maximize the entropy of confidence for each image of the classes to be forgotten. The loss function for the classes to be forgotten is as follows:

$$\mathcal{L}_{\text{forget}}(\mathbf{p}, C) = -\frac{1}{C} \sum_{i=0}^{C-1} \log p_i, \tag{2}$$

where $\mathbf{p}$ and $C$ are the same as Eq. (1).

To summarize, the final objective that we minimize becomes $\mathcal{L} = \mathcal{L}_{\text{memorize}} + \mathcal{L}_{\text{forget}}$. We optimize latent contexts by CMA-ES using the final objective $\mathcal{L}$.

### 3.3 Derivative-Free Optimization: CMA-ES

Since backpropagation cannot be applied to black-box models, we adopt the CMA-ES (Covariance Matrix Adaptation Evolution Strategy) [Hansen et al., 2003], which is a derivative-free optimizer for continuous optimization. In the $t$-th iteration, the CMA-ES samples the $\lambda$ candidate solutions from a multivariate normal distribution $\mathcal{N}(\mathbf{m}_t, \sigma_t^2 \cdot \mathbf{C}_t)$, where $\mathbf{m}_t \in \mathbb{R}^d$ is the mean vector of the search distribution, $\sigma_t \in \mathbb{R}^+$ is the step-size, $\mathbf{C}_t \in \mathbb{R}^{d \times d}$ is a covariance matrix. The $\lambda$ solutions should be evaluated on an objective function $f$, then the CMA-ES updates the parameters $\mathbf{C}_t$, $\mathbf{m}_t$ and $\sigma_t$ by ranking the $\lambda$ solutions by function value (cf. [Hansen, 2016]).

## 4 Experiments

We evaluate the class forgetting performance of our method on image classification tasks. We first describe our experimental setup, including the datasets, baselines, implementation details, and evaluation metrics. We then report the main comparative results between our method and the baselines, as well as a series of analyses of our method.

### 4.1 Setup

**Datasets.** We use four benchmark datasets, i.e., CIFAR-10, CIFAR-100, CUB-200-2011, and ImageNet30. CIFAR-10[3] and CIFAR-100[4] comprise of a total of 50,000 training images and 10,000 test images [Krizhevsky et al., 2009]. These datasets have 10 and 100 classes, respectively. CUB-200-2011 [Wah et al., 2011] comprises of images of 200 distinct bird species, with 5,994 training images and 5,794 test images. ImageNet30 [Hendrycks et al., 2019] is a 30-class subset of the original ImageNet-1k dataset [Deng et al., 2009] (The results on the original ImageNet-1k dataset can also be found in Appendix A.2). It consists of 39,000 training images and 3,000 test images. We conduct experiments in the few-shot condition. We randomly select different $k$ samples of each class from the original training images to construct a $k$-shot training set and a $k$-shot validation set. We set $k$ to 16 for CIFAR-10 and ImageNet30, 4 for CIFAR-100, 1 for CUB-200-2011. For testing, we use the original test set. Unless otherwise noted, the first $40\%$ of classes are to be forgotten, while the other classes are to be memorized.

**Baselines.** Black-Box Forgetting has never been studied before, and there is no existing method directly applicable to this problem. So we compare the proposed method with zero-shot CLIP [Radford et al., 2021], BBT [Sun et al., 2022b] and CBBT [Guo et al., 2023], which are the reasonable baselines as it is for black-box prompt tuning. We apply the same loss functions as our method to BBT and CBBT (w/o adapter) for comparison. CBBT introduces a method that combines textual prompt tuning and adapting output visual features for black-box vision-language models. However, in this paper, the conditions are such that visual features cannot be obtained, and only the final inference results can be used. So we compared the proposed method with CBBT without using any adapter. We also apply the same loss functions as our method to CoOp [Zhou et al., 2022b], a whitebox method, and use the results as reference values under conditions where information on the target model is available.

---

[3,4] `https://www.cs.toronto.edu/~kriz/cifar.html`

Table 1: **Comparisons with the baselines.** The best value is shown in **bold**. BBT [Sun et al., 2022b] and CBBT (w/o adapter) [Guo et al., 2023] are the reasonable baselines as these are for black-box prompt tuning. CoOp [Zhou et al., 2022b] is a white-box method and is included for a reference. Performance is evaluated using the three metrics: the error $Err_{\text{for}}$ for the classes to be forgotten, the accuracy $Acc_{\text{mem}}$ for the classes to be memorized, the harmonic mean $H$ of $Err_{\text{for}}$ and $Acc_{\text{mem}}$. Higher values mean better performance.

| Method | CIFAR-10 | | | CIFAR-100 | | |
|---|---|---|---|---|---|---|
| | $H \uparrow$ | $Err_{\text{for}} \uparrow$ | $Acc_{\text{mem}} \uparrow$ | $H \uparrow$ | $Err_{\text{for}} \uparrow$ | $Acc_{\text{mem}} \uparrow$ |
| Zero-Shot CLIP | 15.30 | 8.37 | 89.05 | 42.14 | 31.17 | 65.03 |
| BBT | $85.69_{\pm0.02}$ | $79.31_{\pm0.03}$ | $93.19_{\pm0.01}$ | $78.36_{\pm0.01}$ | $87.30_{\pm0.01}$ | $71.09_{\pm0.00}$ |
| CBBT | $93.48_{\pm0.02}$ | $90.99_{\pm0.04}$ | $\mathbf{96.11}_{\pm0.00}$ | $73.20_{\pm0.00}$ | $72.69_{\pm0.01}$ | $\mathbf{73.72}_{\pm0.00}$ |
| Ours (w/o LCS) | $72.37_{\pm0.13}$ | $58.57_{\pm0.17}$ | $94.68_{\pm0.01}$ | $79.38_{\pm0.02}$ | $89.17_{\pm0.03}$ | $71.52_{\pm0.01}$ |
| Ours | $\mathbf{95.07}_{\pm0.01}$ | $\mathbf{96.10}_{\pm0.02}$ | $94.06_{\pm0.01}$ | $\mathbf{80.99}_{\pm0.01}$ | $\mathbf{93.37}_{\pm0.02}$ | $71.52_{\pm0.01}$ |
| CoOp (White-Box) | $96.49_{\pm0.00}$ | $96.95_{\pm0.01}$ | $96.04_{\pm0.00}$ | $82.22_{\pm0.00}$ | $99.81_{\pm0.00}$ | $69.90_{\pm0.01}$ |

| Method | CUB-200-2011 | | | ImageNet30 | | |
|---|---|---|---|---|---|---|
| | $H \uparrow$ | $Err_{\text{for}} \uparrow$ | $Acc_{\text{mem}} \uparrow$ | $H \uparrow$ | $Err_{\text{for}} \uparrow$ | $Acc_{\text{mem}} \uparrow$ |
| Zero-Shot CLIP | 46.30 | 46.20 | $\mathbf{46.41}$ | 2.31 | 1.17 | 98.00 |
| BBT | $58.75_{\pm0.01}$ | $88.98_{\pm0.04}$ | $43.85_{\pm0.01}$ | $94.22_{\pm0.05}$ | $90.17_{\pm0.08}$ | $99.06_{\pm0.01}$ |
| CBBT | $56.84_{\pm0.01}$ | $73.52_{\pm0.02}$ | $46.33_{\pm0.01}$ | $87.88_{\pm0.08}$ | $79.69_{\pm0.12}$ | $\mathbf{99.32}_{\pm0.02}$ |
| Ours (w/o LCS) | $58.78_{\pm0.01}$ | $85.85_{\pm0.01}$ | $44.69_{\pm0.01}$ | $95.26_{\pm0.02}$ | $92.19_{\pm0.03}$ | $98.59_{\pm0.01}$ |
| Ours | $\mathbf{59.67}_{\pm0.01}$ | $\mathbf{89.29}_{\pm0.01}$ | $44.81_{\pm0.01}$ | $\mathbf{97.28}_{\pm0.01}$ | $\mathbf{95.94}_{\pm0.01}$ | $98.67_{\pm0.01}$ |
| CoOp (White-Box) | $63.20_{\pm0.02}$ | $98.09_{\pm0.02}$ | $46.62_{\pm0.02}$ | $99.30_{\pm0.01}$ | $99.72_{\pm0.00}$ | $98.89_{\pm0.01}$ |

**Implementation Details.** We use ViT-B/16 [Dosovitskiy et al., 2021] as our CLIP image encoder. We set the number of latent contexts $m = 4$ for CIFAR-10, and $m = 16$ for CIFAR-100, CUB-200-2011 and ImageNet30, respectively. The dimension of a latent context in BBT $d$, Shared Latent Context (SLC) $d^s$, and Unique Latent Contexts (ULC) $d^u$ are set to $d = 10, d^s = 20, d^u = 5$ for CIFAR-10, and $d = 125, d^s = 400, d^u = 100$ for CIFAR-100, CUB-200-2011 and ImageNet30, respectively. For optimization, CMA-ES with the population size of 20 is applied in all the conditions, as done in [Sun et al., 2022b]. We optimize the latent contexts for 400 iterations for CIFAR-10 and ImageNet30, and 800 iterations for CIFAR-100 and CUB-200-2011. All the hyperparameters are tuned on the validation sets, which are distinct from the training and test sets.

**Evaluation Metrics.** We use the following three evaluation metrics: (i) $Err_{\text{for}}$ is the error for the classes to be forgotten; (ii) $Acc_{\text{mem}}$ is the accuracy of the classes to be memorized; (iii) $H$ is the harmonic mean of $Err_{\text{for}}$ and $Acc_{\text{mem}}$ as in [Shibata et al., 2021]. $H$ gives the overall selective forgetting performance as it is a balance between the forgetting rate for the classes to be forgotten and the classification accuracy for the classes to be memorized. Higher values for all these metrics are desirable. For all the experimental results, including those in the Appendix sections, we report the average performance of the three runs with different random seeds, as well as their standard deviations.

## 4.2 Main Results: Comparisons with Baselines

Table 1 shows the main comparative results with the baseline methods. Our method improves zero-shot CLIP and outperforms all the baselines on all the datasets.

Comparing our method with BBT, these two are comparable in $Acc_{\text{mem}}$. However, ours is significantly better than BBT in $Err_{\text{for}}$ and consequently clearly outperforms BBT in $H$ as well for all the datasets; in particular, ours is better than BBT by $9.38\%$ on CIFAR-10. These results suggest that our LCS can optimize the learnable latent contexts more effectively in the black-box setting than BBT, which optimizes all the latent contexts independently.

When comparing our method with CBBT, which is the state-of-the-art black-box tuning method, we can see that ours outperforms CBBT on all the datasets. While ours is slightly inferior in $Acc_{\text{mem}}$, it is significantly better in $Err_{\text{for}}$ on all the datasets. These results demonstrate that our method achieves higher selective forgetting performance than CBBT and is a more suitable for the black-box forgetting task.

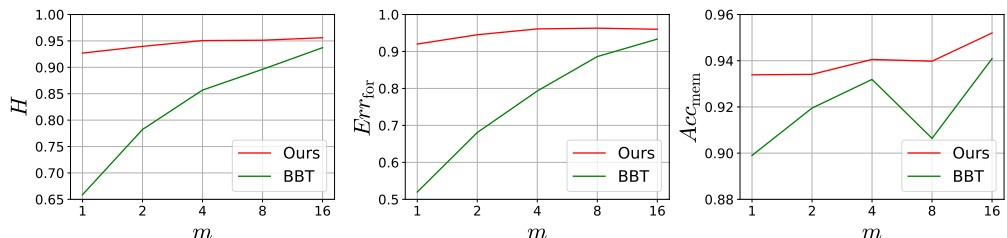

**Figure 3: Sensitivity to the number of latent contexts.** Results of BBT [Sun et al., 2022b] and Ours for varying number of the latent contexts. We can see that our method shows stable performance within a wide range of the number of latent contexts in contrast BBT.

To clarify the impact of our context parametrization method, LCS, we also evaluate the performance of Ours (w/o LCS), i.e., the case without LCS where each latent context is optimized independently. Although these two are highly comparable in $Acc_{\mathrm{mem}}$, Ours (with LCS) clearly outperforms Ours (w/o LCS) by large margins and consequently shows distinct superiority in $H$ as well. These results prove that adequate selective forgetting performance cannot be achieved without LCS. Interestingly, these evaluations clearly suggest that each latent context is not inherently independent and that the idea of LCS to model dependencies among latent contexts by introducing a common latent component is valid.

Finally, we compare our method with the white-box method based on CoOp [Zhou et al., 2022b], where the latent contexts are optimized by minimizing the same loss functions as ours through the standard backpropagation process. Although CoOp is better able to increase $Err_{\mathrm{for}}$ than Ours, the differences are within the acceptable ranges. For example, the difference in $Err_{\mathrm{for}}$ is less than $1\%$ and that in $H$ is less than $2\%$ on CIFAR-10. These results show that even though our method is black-box, it can perform as well as white-box approaches.

### 4.3 Analysis

#### 4.3.1 Sensitivity to The Number of Latent Contexts

We investigate the sensitivity of the performance of our method to the number of latent contexts $m$. Fig. 3 shows $Err_{\mathrm{for}}$, $Acc_{\mathrm{mem}}$ and $H$ when the number of latent contexts $m$ is varied on CIFAR-10. As the number of latent contexts $m$ increases, the performance in all the metrics tends to improve. This is natural behavior, as the performance improves as the number of trainable parameters increases. Comparing Ours and BBT, we see that Ours for $m = 4$ and BBT for $m = 16$ are almost comparable. This means that BBT needs to optimize about four times the number of dimensions to compete with our LCS, indicating that our LCS provides excellent trade-offs. Furthermore, while BBT suffers a sharp drop in accuracy with decreasing $m$, our method shows only a small decrease from $m = 16$ even when $m = 1$. This suggests that our LCS is robust to the decrease in the number of latent contexts and its significant superiority to BBT for latent context representation.

#### 4.3.2 Sensitivity to The Dimensionality of SLC and ULC

We investigate the sensitivity of our method to $d^s$ and $d^u$, which are the number of dimensions of SLC and ULC, respectively. In our LCS, the total number of dimensions $d$ to be optimized is determined as $d = d^s + m \times d^u$ (see Sec. 3.1); we evaluate the performance when we change the ratio $d^s : d^u$ under the condition where $d = 40$ and $m = 4$ for CIFAR-10, and $d = 2000$ and $m = 16$ for CIFAR-100 and CUB-200-2011. Fig. 4 shows the results on the three datasets. First, we can see that for all the datasets, the performance in $Err_{\mathrm{for}}$ and $H$ significantly degrades for $d^s = 0$ (i.e., when no SLC is used) than for the other cases, which supports the validity of the core idea of our LCS that introduces the shared components. As expected, the performance is substantially improved by setting the appropriate ratio. This is a natural trade-off: if $d^s$ (i.e., the number of dimensions of SLC) is too large, the ability to represent the context is reduced and performance deteriorates; conversely, if $d^u$ (i.e., the number of dimensions of ULC) is too large, optimization becomes difficult and performance deteriorates. Meanwhile, our method achieves satisfactory performance in the wide range of $d^s : d^u$. Second, when we compare Ours with BBT, Ours is superior to BBT in the wide range of $d^s : d^u$. These results justify the strong robustness of our method to $d^s$ and $d^u$.

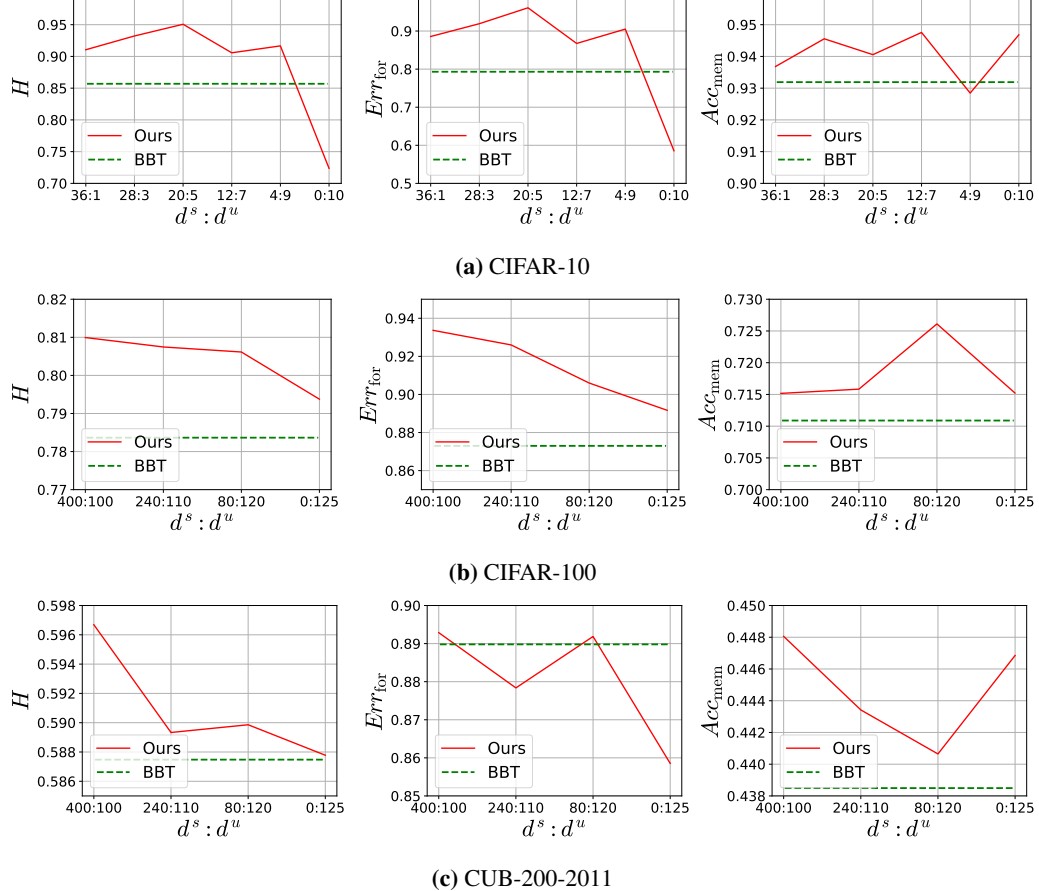

**(a) CIFAR-10**

**(b) CIFAR-100**

**(c) CUB-200-2011**

Figure 4: **Sensitivity to the dimensionality of SLC and ULC.** Results of BBT [Sun et al., 2022b] and Ours for varying number of dimensions of SLC $d^s$ and ULC $d^u$. We can see the effectiveness of our method in the wide range of $d^s : d^u$.

Table 2: **Ours vs. BBT with modified CMA-ES.** Results of VkD-CMA-ES are for $k = 30$ on CIFAR-10 and for $k = 1500$ on CIFAR-100, CUB-200-2011, and ImageNet30.

| Method | CIFAR-10 | | | CIFAR-100 | | |
|---|---|---|---|---|---|---|
| | $H \uparrow$ | $Err_{for} \uparrow$ | $Acc_{mem} \uparrow$ | $H \uparrow$ | $Err_{for} \uparrow$ | $Acc_{mem} \uparrow$ |
| BBT w/ Sep | $93.38_{\pm 0.01}$ | $93.33_{\pm 0.02}$ | $\mathbf{94.64}_{\pm 0.01}$ | $69.29_{\pm 0.01}$ | $67.83_{\pm 0.02}$ | $70.81_{\pm 0.01}$ |
| BBT w/ VkD | $94.11_{\pm 0.00}$ | $95.25_{\pm 0.02}$ | $92.99_{\pm 0.01}$ | $75.41_{\pm 0.02}$ | $79.56_{\pm 0.01}$ | $\mathbf{71.67}_{\pm 0.01}$ |
| Ours | $\mathbf{95.07}_{\pm 0.01}$ | $\mathbf{96.10}_{\pm 0.02}$ | $94.06_{\pm 0.01}$ | $\mathbf{80.99}_{\pm 0.01}$ | $\mathbf{93.37}_{\pm 0.02}$ | $71.52_{\pm 0.01}$ |

| Method | CUB-200-2011 | | | ImageNet30 | | |
|---|---|---|---|---|---|---|
| | $H \uparrow$ | $Err_{for} \uparrow$ | $Acc_{mem} \uparrow$ | $H \uparrow$ | $Err_{for} \uparrow$ | $Acc_{mem} \uparrow$ |
| BBT w/ Sep | $53.74_{\pm 0.02}$ | $74.72_{\pm 0.02}$ | $41.96_{\pm 0.02}$ | $91.18_{\pm 0.02}$ | $84.44_{\pm 0.04}$ | $\mathbf{99.07}_{\pm 0.00}$ |
| BBT w/ VkD | $55.12_{\pm 0.02}$ | $81.49_{\pm 0.03}$ | $41.65_{\pm 0.02}$ | $91.25_{\pm 0.05}$ | $84.58_{\pm 0.09}$ | $99.06_{\pm 0.00}$ |
| Ours | $\mathbf{59.67}_{\pm 0.01}$ | $\mathbf{89.29}_{\pm 0.01}$ | $\mathbf{44.81}_{\pm 0.01}$ | $\mathbf{97.28}_{\pm 0.01}$ | $\mathbf{95.94}_{\pm 0.01}$ | $98.67_{\pm 0.01}$ |

### 4.3.3 Ours vs. BBT with Modified CMA-ES

CMA-ES [Hansen et al., 2003] is widely used in black-box optimization, but when applied to high-dimensional black-box continuous optimization, the computational complexity can become a hindrance. To apply CMA-ES to high-dimensional optimization, modified versions such as Sep-CMA-ES (Sep) [Ros and Hansen, 2008] and VkD-CMA-ES (VkD) [Akimoto and Hansen, 2016] have been developed. Sep-CMA-ES realizes the computational complexity $O(d)$ that is linear with respect to the number of dimensions by restricting the covariance matrix to a diagonal matrix. In other words, the solution generation distribution of Sep-CMA-ES does not consider the covariance between variables, but only the variance of each variable. In VkD-CMA-ES, the covariance matrix

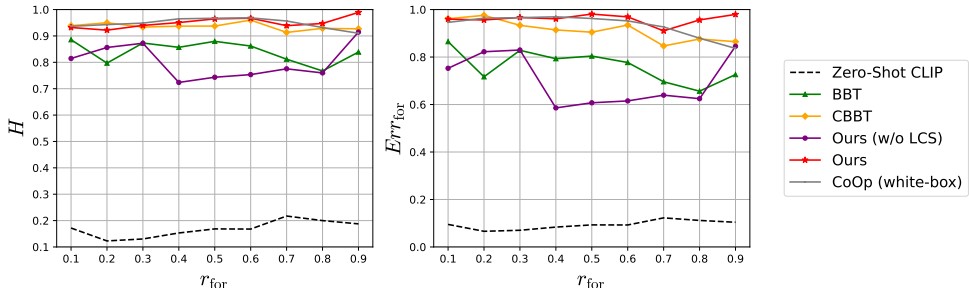

**Figure 5: Sensitivity to the ratio of the classes to be forgotten**. Results on CIFAR-10 when changing the ratio of the classes to be forgotten. Our method is robust to the ratio of the classes to be forgotten compared to baselines.

is expressed as $\mathbf{C} = \mathbf{D}(\mathbf{I}_d + \mathbf{V}\mathbf{V}^T)\mathbf{D}$, where $\mathbf{D}$ is a diagonal matrix, $\mathbf{V}$ is a $d \times k$-dimensional real matrix in which each column is orthogonal, and $k$ is a hyperparameter that determines the degree of freedom of the covariance matrix model. Given the availability of these modified CMA-ESs, one question would be that, can our method still perform better than BBT, even when it is combined with these modified CMA-ESs? Table 2 shows the comparisons between Ours and BBTs with the above two variants of CMA-ES. Ours achieves the best performance in terms of $H$ and $Err_{\text{for}}$ for all the datasets. In particular, the two BBT variants show $Err_{\text{for}}$ more than $10\%$ lower than our method on CIFAR-100 and ImageNet30. These results show that our LCS surpasses Sep-CMA-ES and VkD-CMA-ES as a context parametrization method in the black-box forgetting task.

### 4.3.4 Sensitivity to The Ratio of The Classes To Be Forgotten

Fig. 5 shows $Err_{\text{for}}$ and $H$ when changing the ratio of the classes to be forgotten $r_{\text{for}}$ on CIFAR-10. For BBT, we see a decreasing trend in $Err_{\text{for}}$ as the number of classes to be forgotten increases. This suggests that the context representation of BBT is inefficient, making it difficult to forget multiple classes at a time. CBBT shows some robustness, but as with BBT, we see that $Err_{\text{for}}$ tends to decrease as the number of classes to be forgotten increases. In contrast, our method does not decrease $Err_{\text{for}}$ even if the number of classes to be forgotten increases, which confirms the strong performance of our LCS. In terms of $H$, compared to the baselines, our method shows high robustness independent of the number of classes to be forgotten and achieves high selective forgetting performance.

## 5    Limitations

Our method optimizes the context (token) embeddings of the model through a derivative-free optimization, CMA-ES. That is, we assume that we have access to the context embeddings of the target model. This is a common black-box setting, as similar assumptions have also been made in most of the existing studies [Sun et al., 2022b,a, Guo et al., 2023]. However, there should be models in the real world with a higher level of "black boxness," i.e., models in which even access to contextual embeddedness is prohibited. Addressing such a case is a subject for future research.

## 6    Conclusion

We proposed Black-Box Forgetting, a novel problem of selective forgetting for black-box models. We introduced Latent Context Sharing (LCS), an efficient and effective parametrization method of prompt, which is suitable for derivative-free optimization. Experimental results demonstrated that our method outperforms the reasonable baselines with significant margins. In addition, the sensitivity analyses showed the effectiveness of our LCS.

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

# A Appendix

## A.1 Broader Impacts

In this paper, we introduced the novel problem called Black-Box Forgetting, i.e., the task of selective forgetting for large-scale and black-box pre-trained models, and demonstrated the potential effectiveness of parametrizing the latent contexts in the text prompts into unique and shared components. We here would like to emphasize several potential benefits of our work.

1. Toward addressing the "Right to be Forgotten": It should comply with the request that if a service provider is asked to remove information so that their model cannot recognize certain information. This can be accomplished by training the model from scratch by removing samples of that class from the training data. However, retraining a large-scale model consumes an enormous amount of energy, which should be avoided. Selective forgetting may provide an efficient solution to this problem.

2. Toward efficient large-scale pre-trained models: Improving the efficiency of large-scale pre-trained models is of current interest to many researchers. Various attempts have been made such as model compression and architecture optimization (e.g., `https://sites.google.com/view/elvm/program`). Meanwhile, as the "scaling law" indicates, the reasonable size of a model correlates with the amount of knowledge it memorizes. Therefore, if the number of classes (amount of knowledge) that can be recognized by the model is limited, the model can be scaled down accordingly, thereby improving the efficiency of the model. This contributes to expanding the applicability of large-scale pre-trained models.

3. Toward better control over text-to-image generation: While diffusion-based text-to-image generation can generate diverse types of high-quality images, controlling the content of images remains a challenge. Recent research has focused on "forgetting" visual concepts in order to avoid generating undesirable content [Graves et al., 2021, Lu et al., 2024, Zhang et al., 2024]. These methods forget certain classes by directly fine-tuning the diffusion model, but tuning the diffusion model itself is often costly. In contrast, given that typical generative models use a text encoder of a pre-trained VLM (e.g., CLIP) for conditioning, our method may provide an efficient approach to class forgetting by fine-tuning only the prompts of the text encoder.

We believe that our paper will open new directions for these important problems of interest to the machine learning community, even if these are not immediately feasible with this paper alone.

## A.2 Comparative Results on ImageNet-1k

We conduct experiments on ImageNet-1k [Deng et al., 2009] to verify the effectiveness of our method on a larger-scale dataset. The setup is the same as the cases for CIFAR-100 and CUB-200-2011 (see Sec. 4.1). The results are shown in Table 3. Our method outperforms all the baselines in terms of $H$ and $Err_{\text{for}}$, which supports the effectiveness of our method further.

**Table 3: Comparative results on ImageNet-1K.**

| Method | $H \uparrow$ | $Err_{\text{for}} \uparrow$ | $Acc_{\text{mem}} \uparrow$ |
|---|---|---|---|
| Zero-Shot CLIP | 41.79 | 30.78 | 65.07 |
| BBT | $75.08_{\pm 0.01}$ | $93.58_{\pm 0.02}$ | $62.68_{\pm 0.00}$ |
| CBBT | $75.28_{\pm 0.01}$ | $88.53_{\pm 0.03}$ | $\mathbf{65.49}_{\pm 0.00}$ |
| Ours (w/o LCS) | $74.65_{\pm 0.00}$ | $91.73_{\pm 0.01}$ | $62.94_{\pm 0.00}$ |
| Ours | $\mathbf{76.35}_{\pm 0.00}$ | $\mathbf{94.80}_{\pm 0.01}$ | $63.91_{\pm 0.00}$ |
| CoOp (White-Box) | $77.42_{\pm 0.01}$ | $94.78_{\pm 0.03}$ | $65.43_{\pm 0.01}$ |

## A.3 Trade-off between $Err_{\text{for}}$ and $Acc_{\text{mem}}$

Tables 1 and 3 shows that our method performs poorly in $Acc_{\text{mem}}$ than the baseline methods. This is because $Err_{\text{for}}$ and $Acc_{\text{mem}}$ are in a trade-off relationship, with $Acc_{\text{mem}}$ tending to decrease as $Err_{\text{for}}$ is increased. This is presumably because features between classes are not completely

disentangled in the feature space, so forgetting one class may negatively affect other classes (just as dog and cat share some common features). To provide justification for this, we report the results of using a loss more prioritized (weighted) for $Acc_{\text{mem}}$ in our method in Table 4 as "Ours ($Acc$ prio.)." We can see that Ours ($Acc$ prio.) outperforms all the other methods in $Acc_{\text{mem}}$, with sacrificing $Err_{\text{for}}$. Notably, both Ours and Ours ($Acc$ prio.) outperform BBT and CBBT in $H$, indicating that our method achieves a better trade-off than BBT and CBBT.

**Table 4: Trade-off between $Err_{\textbf{for}}$ and $Acc_{\textbf{mem}}$ evaluated on CUB-200-2011.**

| Method | $H \uparrow$ | $Err_{\text{for}} \uparrow$ | $Acc_{\text{mem}} \uparrow$ |
|---|---|---|---|
| Zero-Shot CLIP | 46.30 | 46.20 | 46.41 |
| BBT | $58.75_{\pm 0.01}$ | $88.98_{\pm 0.04}$ | $43.85_{\pm 0.01}$ |
| CBBT | $56.84_{\pm 0.01}$ | $73.52_{\pm 0.02}$ | $46.33_{\pm 0.01}$ |
| Ours (w/o LCS) | $58.78_{\pm 0.01}$ | $85.85_{\pm 0.01}$ | $44.69_{\pm 0.01}$ |
| Ours | $\mathbf{59.67}_{\pm 0.01}$ | $\mathbf{89.29}_{\pm 0.01}$ | $44.81_{\pm 0.01}$ |
| Ours ($Acc$ prio.) | $59.17_{\pm 0.00}$ | $80.87_{\pm 0.00}$ | $\mathbf{46.65}_{\pm 0.01}$ |
| CoOp | $63.20_{\pm 0.02}$ | $98.09_{\pm 0.02}$ | $46.62_{\pm 0.02}$ |

Furthermore, the results on CUB-200-2011 in Table 1 show that the white-box method CoOp [Zhou et al., 2022b] only improves $Acc_{\text{mem}}$ by 0.2% from zero-shot and our method even hurts $Acc_{\text{mem}}$. Our problem is to achieve forgetting only specified classes while maintaining the accuracy of the other classes to be memorized, i.e., to improve $Err_{\text{for}}$ while maintaining $Acc_{\text{mem}}$. Achieving both of these is more challenging than merely improving $Acc_{\text{mem}}$ alone. To verify our argument here, we report an analysis in Table 5. "CoOp (White-Box w/ only memorization)" shows the results when white-box tuning by CoOp is performed to minimize the cross-entropy loss over only the classes to be memorized, i.e., forgetting is not performed. We can see a significant improvement in $Acc_{\text{mem}}$. This result proves that, even in a white-box setting, achieving both improving $Acc_{\text{mem}}$ and $Err_{\text{for}}$ is more difficult than merely improving $Acc_{\text{mem}}$ alone. Since the black-box setting is generally more challenging than the white-box setting, it is not at all surprising that our method leads to a slight degradation in $Acc_{\text{mem}}$.

**Table 5: White-box CoOp with only memorization on CUB-200-2011.**

| Method | $H \uparrow$ | $Err_{\text{for}} \uparrow$ | $Acc_{\text{mem}} \uparrow$ |
|---|---|---|---|
| Zero-Shot CLIP | 46.30 | 46.20 | 46.41 |
| Ours | $59.67_{\pm 0.01}$ | $89.29_{\pm 0.01}$ | $44.81_{\pm 0.01}$ |
| CoOp (White-Box) | $63.20_{\pm 0.02}$ | $98.09_{\pm 0.02}$ | $46.62_{\pm 0.02}$ |
| CoOp (White-Box w/ only memorization) | $48.94_{\pm 0.00}$ | $44.01_{\pm 0.01}$ | $55.10_{\pm 0.02}$ |

### A.4 On Zero-shot Approach to Forgetting

In this paper, we consider a few-shot learning scenario, i.e., we can access a small number of training examples for tuning the latent contexts. However, in some practical cases, one might be faced with a zero-shot scenario, i.e., a situation where no sample is available. We here consider a simple zero-shot tuning approach to the black-box forgetting task by only using the class names (i.e., class embeddings). Specifically, let $\mathbf{z}_c$ and $\mathbf{z}$ denote the class embeddings before and after prompt tuning for the class to be forgotten, respectively. We assume only $\mathbf{z}$ is trainable and aim to tune $\mathbf{z}$ by minimizing the following negative NT-Xent loss:

$$\mathcal{L}_{\text{NT}-\text{Xent}} = \log \frac{\exp(\mathbf{z}_c^\top \mathbf{z}/\tau)}{\sum_i \exp(\mathbf{z}_i^\top \mathbf{z}/\tau)}, \tag{3}$$

where $\mathbf{z}_i$ is the class embedding for the $i$-th class to be kept memorized. This loss requires $\mathbf{z}$ to be orthogonal to $\mathbf{z}_c$ as well as be similar to the embeddings of the other classes $\{\mathbf{z}_i\}$. By minimizing the loss, we can expect that the embedding of the class to be forgotten $\mathbf{z}_c$ is placed equidistant from the embeddings of all the other classes $\{\mathbf{z}_i\}$.

The results are shown in Table 6. We found that the zero-shot approach (C-Emb.) performs significantly poorly compared to our few-shot learning approach. A closer look at each evaluation metric shows a slight advantage of C-Emb. over Ours in $Err_{\text{for}}$, but C-Emb. is far lower than Ours in $Acc_{\text{mem}}$. This is probably due to the nature of the negative NT-Xent loss, whereby the embedding

of the class to be forgotten is closer to the embeddings of the other classes to be memorized, resulting in more misclassifications. These results prove that tuning with only the class embeddings does not provide satisfactory performance.

**Table 6: Zero-shot (C-Emb.) vs. Few-shot (Ours).**

| Method | CIFAR-10 | | | CIFAR-100 | | |
|---|---|---|---|---|---|---|
| | $H \uparrow$ | $Err_{\text{for}} \uparrow$ | $Acc_{\text{mem}} \uparrow$ | $H \uparrow$ | $Err_{\text{for}} \uparrow$ | $Acc_{\text{mem}} \uparrow$ |
| C-Emb. | $91.83_{\pm 0.01}$ | $\mathbf{99.04_{\pm 0.01}}$ | $85.61_{\pm 0.01}$ | $64.00_{\pm 0.01}$ | $\mathbf{99.68_{\pm 0.00}}$ | $47.14_{\pm 0.02}$ |
| Ours | $\mathbf{95.07_{\pm 0.01}}$ | $96.10_{\pm 0.02}$ | $\mathbf{94.06_{\pm 0.01}}$ | $\mathbf{80.99_{\pm 0.01}}$ | $93.37_{\pm 0.02}$ | $\mathbf{71.52_{\pm 0.01}}$ |

| Method | CUB-200-2011 | | | ImageNet30 | | |
|---|---|---|---|---|---|---|
| | $H \uparrow$ | $Err_{\text{for}} \uparrow$ | $Acc_{\text{mem}} \uparrow$ | $H \uparrow$ | $Err_{\text{for}} \uparrow$ | $Acc_{\text{mem}} \uparrow$ |
| C-Emb. | $51.83_{\pm 0.05}$ | $\mathbf{99.79_{\pm 0.00}}$ | $35.13_{\pm 0.04}$ | $67.47_{\pm 0.18}$ | $90.72_{\pm 0.13}$ | $58.04_{\pm 0.22}$ |
| Ours | $\mathbf{59.67_{\pm 0.01}}$ | $89.29_{\pm 0.01}$ | $\mathbf{44.81_{\pm 0.01}}$ | $\mathbf{97.28_{\pm 0.01}}$ | $\mathbf{95.94_{\pm 0.01}}$ | $\mathbf{98.67_{\pm 0.01}}$ |

The above results give us further inspiration; If training samples are available for some classes and not for the rest, would there be a benefit from combining the two approaches? We experimented under conditions in which half of the classes to be forgotten had training samples available and the rest did not. We use Ours for the classes for which the training samples are available and C-Emb. for the classes with no training samples. The result on ImageNet30 are shown in Table 7. While Ours performed forgetting for only the classes with the training samples based on the loss given in Sec. 3.2, Ours + C-Emb. performed forgetting for all the classes by incorporating Eq. 3. We see that Ours + C-Emb. could outperform Ours in all the metrics, which proves the effectiveness of the combination.

**Table 7: Combining Zero-shot and Few-shot.**
Ours + C-Emb. applies our few-shot approach to only the classes for which the training samples are avilable and C-Emb. to those with no training samples.

| Method | $H \uparrow$ | $Err_{\text{for}} \uparrow$ | $Acc_{\text{mem}} \uparrow$ |
|---|---|---|---|
| Ours + C-Emb. | $\mathbf{92.30_{\pm 0.05}}$ | $\mathbf{86.56_{\pm 0.08}}$ | $\mathbf{98.87_{\pm 0.00}}$ |
| Ours | $89.34_{\pm 0.03}$ | $81.56_{\pm 0.05}$ | $98.78_{\pm 0.00}$ |

## A.5   Forgetting vs. Learning from Scratch

An obvious alternative to forgetting that produces a model that can recognize only the classes to be memorized while preventing the recognition of only the classes to be forgotten is to learn the model from scratch for only the classes to be memorized. To clarify the benefit of forgetting, we evaluate the accuracy of ResNet-18 and ViT-B/16 trained from scratch over only the classes to be memorized. The results are shown in Table 8. As can be seen, both models cause severe overfitting to the training data. That is, while these models achieve reasonable accuracy on the training data, they exhibit severely poor performance on the test data. We also tested both models when initialized with ImageNet pre-trained weights. While the results are improved to some extent for ResNet-18, these are still far behind our forgetting method. The reason for this is that, following the common protocol in context optimization [Zhou et al., 2022b], we conducted our experiments in few-shot scenarios as explained in Sec. 4.1, which overwhelmingly lacks the number of training samples to learn the weights for even ResNet-18 and Vit-B/16. These results demonstrate the superiority of our forgetting approach, which achieves effective tuning with a small sample size.

## A.6   Forgetting vs. Fine-tuning over The Classes To Be Memorized

Another alternative approach to forgetting would be to only fine-tuning the model over the classes to be memorized. That is, do nothing to the classes to be forgotten and cause catastrophic forgetting. We test the performance of fine-tuned CLIP over only the classes to be memorized (Fine-tune CLIP). The results are shown in Table 9. Compared to our method, which explicitly facilitates forgetting of the classes to be forgotten, catastrophic forgetting alone (Fine-tune CLIP) is not sufficient to achieve satisfactory forgetting performance. Also, our method is comparable to the Fine-tune CLIP in $Acc_{\text{mem}}$ except for CUB-200-2011, which does not significantly impair the classification accuracy for the classes to be memorized. These results support the validity of our forgetting approach.

**Table 8: Forgetting vs. Learning from scratch evaluated in $Acc_{mem} \uparrow$ (ViT-B/16, ResNet18).**

| Method | CIFAR-10 | | CIFAR-100 | |
| --- | --- | --- | --- | --- |
| | Test | Train | Test | Train |
| ResNet18 From Scratch | $39.94_{\pm0.02}$ | $98.96_{\pm0.01}$ | $11.16_{\pm0.00}$ | $96.39_{\pm0.02}$ |
| ViT-B/16 From Scratch | $29.76_{\pm0.00}$ | $68.05_{\pm0.08}$ | $5.53_{\pm0.00}$ | $36.81_{\pm0.08}$ |
| ResNet18 w/ pretrained weights | $60.53_{\pm0.04}$ | $99.65_{\pm0.00}$ | $25.98_{\pm0.01}$ | $100.00_{\pm0.00}$ |
| ViT-B/16 w/ pretrained weights | $34.48_{\pm0.04}$ | $73.96_{\pm0.22}$ | $8.44_{\pm0.00}$ | $91.53_{\pm0.05}$ |
| Ours | $94.06_{\pm0.01}$ | $95.92_{\pm0.00}$ | $71.52_{\pm0.01}$ | $78.61_{\pm0.01}$ |

| Method | CUB-200-2011 | | ImageNet30 | |
| --- | --- | --- | --- | --- |
| | Test | Train | Test | Trainn |
| ResNet18 From Scratch | $1.84_{\pm0.02}$ | $98.89_{\pm0.00}$ | $36.20_{\pm0.01}$ | $98.26_{\pm0.01}$ |
| ViT-B/16 From Scratch | $1.42_{\pm0.00}$ | $52.78_{\pm0.17}$ | $16.56_{\pm0.02}$ | $34.14_{\pm0.06}$ |
| ResNet18 w/ pretrained weights | $7.60_{\pm0.01}$ | $100.00_{\pm0.00}$ | $67.00_{\pm0.03}$ | $99.65_{\pm0.00}$ |
| ViT-B/16 w/ pretrained weights | $1.45_{\pm0.00}$ | $76.11_{\pm0.21}$ | $27.41_{\pm0.05}$ | $98.15_{\pm0.01}$ |
| Ours | $44.81_{\pm0.01}$ | $53.61_{\pm0.01}$ | $98.67_{\pm0.01}$ | $99.19_{\pm0.00}$ |

**Table 9: Forgetting vs. Fine-tuned CLIP over the classes to be memorized.**

| Method | CIFAR-10 | | | CIFAR-100 | | |
| --- | --- | --- | --- | --- | --- | --- |
| | $H \uparrow$ | $Err_{for} \uparrow$ | $Acc_{mem} \uparrow$ | $H \uparrow$ | $Err_{for} \uparrow$ | $Acc_{mem} \uparrow$ |
| Fine-tune CLIP | $59.48_{\pm0.11}$ | $44.21_{\pm0.11}$ | $\mathbf{95.48}_{\pm0.01}$ | $40.89_{\pm0.00}$ | $28.66_{\pm0.00}$ | $71.32_{\pm0.00}$ |
| Ours | $\mathbf{95.07}_{\pm0.01}$ | $\mathbf{96.10}_{\pm0.02}$ | $94.06_{\pm0.01}$ | $\mathbf{80.99}_{\pm0.01}$ | $\mathbf{93.37}_{\pm0.02}$ | $\mathbf{71.52}_{\pm0.01}$ |

| Method | CUB-200-2011 | | | ImageNet30 | | |
| --- | --- | --- | --- | --- | --- | --- |
| | $H \uparrow$ | $Err_{for} \uparrow$ | $Acc_{mem} \uparrow$ | $H \uparrow$ | $Err_{for} \uparrow$ | $Acc_{mem} \uparrow$ |
| Fine-tune CLIP | $48.94_{\pm0.00}$ | $44.01_{\pm0.01}$ | $\mathbf{55.10}_{\pm0.02}$ | $2.68_{\pm0.01}$ | $1.36_{\pm0.01}$ | $98.56_{\pm0.03}$ |
| Ours | $\mathbf{59.67}_{\pm0.01}$ | $\mathbf{89.29}_{\pm0.01}$ | $44.81_{\pm0.01}$ | $\mathbf{97.28}_{\pm0.01}$ | $\mathbf{95.94}_{\pm0.01}$ | $\mathbf{98.67}_{\pm0.01}$ |

## A.7 CoOp with Model Parameter Update

We evaluate another white-box forgetting approach that learns latent contexts through CoOp while also updating the model parameters. The results are shown in Table 10. We see that simultaneously updating the model parameters does not improve performance, but rather hurts it. This is not surprising, as it is known that straightforward fine-tuning of the zero-shot CLIP does not improve performance [Wortsman et al., 2022].

**Table 10: Results of CoOp with updating the model parameters.**

| Method | CIFAR-10 | | | CIFAR-100 | | |
| --- | --- | --- | --- | --- | --- | --- |
| | $H \uparrow$ | $Err_{for} \uparrow$ | $Acc_{mem} \uparrow$ | $H \uparrow$ | $Err_{for} \uparrow$ | $Acc_{mem} \uparrow$ |
| CoOp (White-box) + Parameter Update | $75.74_{\pm0.24}$ | $92.25_{\pm0.05}$ | $73.54_{\pm0.33}$ | $79.92_{\pm0.02}$ | $85.80_{\pm0.04}$ | $74.83_{\pm0.01}$ |
| CoOp (White-box) | $96.49_{\pm0.00}$ | $96.95_{\pm0.01}$ | $96.04_{\pm0.00}$ | $82.22_{\pm0.00}$ | $99.81_{\pm0.00}$ | $69.90_{\pm0.01}$ |
| Ours | $95.07_{\pm0.01}$ | $96.10_{\pm0.02}$ | $94.06_{\pm0.01}$ | $80.99_{\pm0.01}$ | $93.37_{\pm0.02}$ | $71.52_{\pm0.01}$ |

| Method | CUB-200-2011 | | | ImageNet30 | | |
| --- | --- | --- | --- | --- | --- | --- |
| | $H \uparrow$ | $Err_{for} \uparrow$ | $Acc_{mem} \uparrow$ | $H \uparrow$ | $Err_{for} \uparrow$ | $Acc_{mem} \uparrow$ |
| CoOp (White-box) + Parameter Update | $59.78_{\pm0.01}$ | $98.48_{\pm0.01}$ | $42.93_{\pm0.01}$ | $99.29_{\pm0.00}$ | $99.72_{\pm0.00}$ | $98.85_{\pm0.01}$ |
| CoOp (White-box) | $63.20_{\pm0.02}$ | $98.09_{\pm0.02}$ | $46.62_{\pm0.02}$ | $99.30_{\pm0.01}$ | $99.72_{\pm0.00}$ | $98.89_{\pm0.01}$ |
| Ours | $59.67_{\pm0.01}$ | $89.29_{\pm0.01}$ | $44.81_{\pm0.01}$ | $97.28_{\pm0.01}$ | $95.94_{\pm0.01}$ | $98.67_{\pm0.01}$ |

## A.8 Impact of Sampling Distribution for Random Projection

As explained in Sec 3.1, each of the $(d^u + d^s)$-dimensional latent context is projected into the original $D$-dimensional context by a random projection $A \in \mathbb{R}^{D \times (d^u + d^s)}$ sampled from a normal distribution $\mathcal{N}(0, \sigma)$, where $\sigma$ is the standard deviation of the context embeddings. We analyze the impact of the choice of $\sigma$ on the final performance. Table 11 shows the results on CIFAR-10 when

$\sigma$ is fixed to 1. We can see that the choice of $\sigma$ has a significant impact on the performance in all the evaluation metrics.

**Table 11: Impact of sampling distribution for random projection.** Results on CIFAR-10 when we use $\mathcal{N}(0,1)$ and $\mathcal{N}(0,\sigma)$ for drawing the random projection $A$.

| Distribution | $H\uparrow$ | $Err_{\text{for}}\uparrow$ | $Acc_{\text{mem}}\uparrow$ |
|---|---|---|---|
| $\mathcal{N}(0,1)$ | $84.58_{\pm0.05}$ | $78.33_{\pm0.07}$ | $91.91_{\pm0.04}$ |
| $\mathcal{N}(0,\sigma)$ | $\mathbf{95.07}_{\pm0.01}$ | $\mathbf{96.10}_{\pm0.02}$ | $\mathbf{94.06}_{\pm0.01}$ |

### A.9 Performance for Different Choices of The Classes To Be Forgotten

In the experiments in the main paper (Sec. 4), 40% of the classes in each dataset were selected as the classes to be forgotten, in ascending order of the class index, starting with the 0-th class. We here investigate the performance of our method when using different selections of the classes to be forgotten. The results are in Table 12. We see that our method achieves satisfactory selective forgetting performance regardless of the choice of the classes to be forgotten.

**Table 12: Results for different choices of the classes to be forgotten on CIFAR-10.**

| Class indices to be forgotten | $H$ | $Err_{\text{for}}$ | $Acc_{\text{mem}}$ |
|---|---|---|---|
| $\{1\}$ | $94.25_{\pm0.01}$ | $98.33_{\pm0.02}$ | $90.49_{\pm0.01}$ |
| $\{2\}$ | $92.16_{\pm0.04}$ | $93.10_{\pm0.07}$ | $91.24_{\pm0.01}$ |
| $\{0,8\}$ | $94.01_{\pm0.00}$ | $97.67_{\pm0.01}$ | $90.61_{\pm0.00}$ |
| $\{2,5\}$ | $94.48_{\pm0.01}$ | $96.45_{\pm0.03}$ | $92.58_{\pm0.00}$ |
| $\{2,3,4\}$ | $94.41_{\pm0.01}$ | $95.90_{\pm0.01}$ | $92.97_{\pm0.00}$ |
| $\{4,5,6\}$ | $95.53_{\pm0.01}$ | $96.89_{\pm0.02}$ | $94.21_{\pm0.01}$ |
| $\{1,2,3,4\}$ | $93.27_{\pm0.04}$ | $92.74_{\pm0.09}$ | $93.79_{\pm0.02}$ |
| $\{4,5,6,7\}$ | $96.45_{\pm0.01}$ | $99.52_{\pm0.00}$ | $93.57_{\pm0.01}$ |
| $\{3,4,5,6,7\}$ | $95.43_{\pm0.02}$ | $98.33_{\pm0.00}$ | $92.69_{\pm0.03}$ |
| $\{4,5,6,7,8\}$ | $96.17_{\pm0.01}$ | $97.42_{\pm0.02}$ | $94.95_{\pm0.01}$ |
| $\{3,4,5,6,7,8\}$ | $95.15_{\pm0.01}$ | $96.97_{\pm0.03}$ | $93.40_{\pm0.02}$ |
| $\{4,5,6,7,8,9\}$ | $95.57_{\pm0.02}$ | $95.22_{\pm0.03}$ | $95.92_{\pm0.01}$ |
| $\{1,2,3,4,5,6,7\}$ | $94.04_{\pm0.02}$ | $92.40_{\pm0.05}$ | $95.74_{\pm0.01}$ |
| $\{2,3,4,5,6,7,8\}$ | $97.16_{\pm0.00}$ | $98.78_{\pm0.01}$ | $95.60_{\pm0.00}$ |
| $\{1,2,3,4,5,6,7,8\}$ | $97.50_{\pm0.00}$ | $97.69_{\pm0.02}$ | $97.32_{\pm0.02}$ |
| $\{2,3,4,5,6,7,8,9\}$ | $97.94_{\pm0.01}$ | $97.76_{\pm0.01}$ | $98.12_{\pm0.02}$ |
| $\{1,2,3,4,5,6,7,8,9\}$ | $99.51_{\pm0.00}$ | $99.09_{\pm0.01}$ | $99.93_{\pm0.00}$ |
| $\{0,2,3,4,5,6,7,8,9\}$ | $98.83_{\pm0.02}$ | $98.55_{\pm0.02}$ | $99.10_{\pm0.01}$ |

### A.10 Computational Complexity

We use a single GV100 GPU with 12.885GB memory for all the experiments. Approximate computation time is 120 minutes for CIFAR-10, 400 minutes for ImageNet30 and 600 minutes for CIFAR-100 and CUB-200-2011.

