# OpenReview forum: "Black-Box Forgetting"
_NeurIPS.cc/2024/Conference — NeurIPS 2024 poster_

### Official Review · Reviewer_SKod · 2024-07-09

**Soundness:** 3
**Presentation:** 2
**Contribution:** 2
**Rating:** 5
**Confidence:** 4

**Summary:**

This paper explores a selective memorization problem of classification models under a black-box setup. The proposed method selectively applies different learning objectives, cross-entropy minimization, and entropy maximization for classes to be memorized and classes to be unmemorized, respectively; they become guidance for a small amount of text embedding tokens (prompt) by a black-box optimization algorithm, CMA-ES. The authors evaluate the proposed method in terms of selective memorization capability.

**Strengths:**

- The problem setup "black-box forgetting" that the authors focus on is crucial yet unexplored. The authors timely provide a simple approach to address the problem.
- The design of the proposed method is easy to follow and reasonable.

**Weaknesses:**

# 1. Lack of novelty
* The proposed method can be viewed as a variant of BBT [1], which explicitly separates the seed latent context into shared parts and unique parts. Although this is a new application of BBT in selective memorization scenarios, if the authors want to claim the novelty of the modified BBT method, I think they should further provide analyses on efficiency and qualitative results on learned context to contrast with BBT.
* Moreover, the proposed learning objective -a combination of cross-entropy minimization and entropy maximization- is also quite common [2], so it is hard to claim its originality from this work.

# 2. Inappropriate validation setup
* As the main goal of this study is selective memorization of classification models, I expected existing methods for selective memorization and unlearning to be the natural baselines. However, all the comparisons are conducted by training the black-box tuning model via the combination of cross-entropy and entropy. Comparing the proposed method with other available unlearning objectives would be necessary to measure its validity.
* The scope of validation is also limited to three natural object recognition. It is important to validate the methods across a more diverse range of image domains to robustify the conclusion.

---
> Reference
* [1] Black-Box Tuning for Language-Model-as-a-Service, Sun et al. 2022
* [2] Mitigating Information Leakage in Image Representations: A Maximum Entropy Approach, Roy and Boddeti 2019

**Questions:**

- Is the sign of $L_{uniform}$ term in the final objective a typo? To maximize entropy, I think we should minimize $-L_{uniform}$ to maximize entropy.
- For selective memorization, I believe that not only the unmemorization for the selected classes but also the memorization for the remaining classes is crucial. However, in Table 1 CUB-200-2011 results indicate that the white-box method only improves $Acc_{mem}$ 0.2 % from zero-shot, which is a very trivial amount given that the strong few-shot learning capability of CoOp [3] and CoCoOp [4] on the similarly fine-grain visual recognition tasks.  Furthermore, the proposed method even hurts the $Acc_{mem}$ from the zero-shot model. Could you explain why the proposed learning objective shows poor learning capability on the CUB dataset compared to the CIFAR datasets?

---
> Reference
* [3] Learning to Prompt for Vision-Language Models, Zhou et al. 2022
* [4] Conditional Prompt Learning for Vision-Language Models, Zhou et al. 2022

**Limitations:**

The authors adequately stated the limitation in the last section.

---

> ### Author Rebuttal · Authors · 2024-08-06
>
> ### Q1. Lack of novelty.
> Let us recap the main novelties of this paper.
>
> 1. We proposed a novel task called Black-Box Forgetting, which aims to achieve selective class forgetting under the assumption that the parameters and gradients of the model are inaccessible.
>
> 2. Aiming at improving derivative-free prompt tuning for Black-Box Forgetting, we proposed a novel context parametrization method called Latent Context Sharing (LCS) that explicitly models shared and unique components over multiple latent contexts.
>
> To our knowledge, there is no literature addressing Black-Box Forgetting, nor has the idea of explicitly modeling the shared and unique components of learnable contexts in prompt tuning been explored. Moreover, in all the experiments reported in our paper, we compared our LCS with BBT and showed that ours was superior to BBT. **We would be happy to discuss these novelties in more detail if the reviewer could provide specific examples of publications that contradict these novelties.**
>
> ### Q2. Loss function not novel.
> Note that we are not claiming novelty in the loss function. We argue that our novelty lies in Latent Context Sharing (LCS), a novel parametrization of the learnable latent contexts in prompt tuning for Black-Box Forgetting.
>
> ### Q3. Baselines are black-box tuning methods, not forgetting methods.
> As far as we know, there is no existing method for Black-Box Forgetting, i.e., there is no existing forgetting / unlearning method that can be directly compared. Therefore, we compared our method with the existing black-box tuning methods, i.e., BBT and CBBT, under the same loss function. **We would appreciate it if the reviewer could provide any specific existing methods that are directly comparable or existing loss functions to be applied so that we can have clearer discussions.**
>
> ### Q4. Datasets limited to three natural image datasets.
> Due to the nature of the pre-trained CLIP that are naturally biased toward natural images, the performance of zero-shot CLIP has been reported to be significantly degraded in non-natural image domains (e.g., [a,b]). Since our task is selective forgetting of classes that can successfully be recognized by the pre-trained model, it is difficult to test its effectiveness on non-natural image datasets.
>
> On the other hand, we agree that conducting experiments on more diverse datasets is beneficial, so we conducted additional experiments using ImageNet30. The results can be found in Table G in the attached PDF. Our method is superior to the other methods in terms of $H$, which supports the effectiveness of our method further.
>
> [a] Shu et al., Test-Time Prompt Tuning for Zero-Shot Generalization in Vision-Language Models, in Proc. NeurIPS, 2022.
>
> [b] Gondal et al., Domain Aligned CLIP for Few-shot Classification, in Proc. WACV, 2024.
>
> ### Q4. Typo in $L_{uniform}$.
>
> We will fix it in the final version. Thanks!
>
> ### Q5. CUB-200-2011 in Table 1, white-box CoOp only improves $Acc_\text{mem}$ by 0.2%, and the proposed method even hurts $Acc_\text{mem}$.
>
> We respectfully disagree with the reviewer's premise. Our problem is to achieve forgetting only specified classes while maintaining the accuracy of the other classes to be memorized, i.e., to improve $Err_\text{for}$ while maintaining $Acc_\text{mem}$. Achieving both of these is more challenging than merely improving $Acc_\text{mem}$ alone.
>
> To verify our argument here, we report an additional quick analysis in Table H in the attached PDF. "CoOp (White-Box w/ only memorization)" shows the results when white-box tuning by CoOp is performed to minimize the cross-entropy loss over only the classes to be memorized, i.e., forgetting is not performed. We can see a significant improvement in $Acc_\text{mem}$, as expected by the reviewer. This result proves that, even in a white-box setting, achieving both improving $Acc_\text{mem}$ and $Err_\text{for}$ is more difficult than merely improving $Acc_\text{mem}$ alone. Since the black-box setting is generally more challenging than the white-box setting, it is not at all surprising that our method leads to a slight degradation in $Acc_\text{mem}$.

---

> ### Comment · Reviewer_SKod · 2024-08-08
>
> I appreciate the authors' comprehensive responses!
> Most of my concerns are well-addressed.
>
> About the novelty and contribution, I think if the authors want to claim the novelty of this work on the problem setup itself, i.e., Black-Box Forgetting, this problem should be challenging so that naive application of the existing method is struggled with.
>
> For example, similar work on the Black-Box optimization setup [Oh et al. 2023] clarifies the problems of naive extension of the white-box methods [Bahng et al. 2022] and addresses those problems with their new method.
> However, in Table 2. of the submitted draft, **BBT with a different configuration of CMA-ES, which is a naive extension of the existing method, already achieves very strong performance**. This raises doubt about the difficulty of the considered problem setup, and given that, I still lean towards the negative side because the proposed method, LCS, lacks the necessity.
>
> Could the authors give further comment and opinion on this?
>
> ---
>
> - Oh et al. 2023, BlackVIP: Black-Box Visual Prompting for Robust Transfer Learning
> - Bahng et al. 2022, Exploring Visual Prompts for Adapting Large-Scale Models

---

> > ### Author Response · Authors · 2024-08-11
> > **Further response to the question of the novelty of the Black-Box Forgetting problem**
> >
> > Thank you for taking the time to read through our rebuttal so promptly! We are very pleased to hear that most of your questions have been properly addressed.
> >
> > Thank you also for clarifying your question about the novelty of our problem, Black-Box Forgetting. However, we would like to argue that novelty of a problem should not necessarily be judged based on the difference in performance between existing and proposed methods. In fact, looking back at the history of machine learning, even if the difference in accuracy between existing and proposed solutions is less than 5%, there are many studies that have brought about significant advances in community by introducing novel problems. For example, few-shot object detection [a], universal domain adaptation [b], and open-set semi-supervised learning [c], just to name a few (You would agree that it is not possible to determine a golden rule like "If and only if there is more than $x$% difference between the proposed and existing methods, the problem will be considered novel.").
> >
> > We also want to argue that our Black-Box Forgetting is challenging. Besides Table 2 on CIFAR-10, below we report results for three other more challenging datasets.
> >
> > | | |CIFAR-100| | |CUB-200-2011| | |ImageNet30| |
> > |:-------|:-------:|:-------:|:-------:|:-------:|:-------:|:-------:|:-------:|:-------:|:-------:|
> > |Method|$H\uparrow$|$Err_{\mathrm{for}}\uparrow$|$Acc_{\mathrm{mem}}\uparrow$|$H\uparrow$|$Err_{\mathrm{for}}\uparrow$|$Acc_{\mathrm{mem}}\uparrow$|$H\uparrow$|$Err_{\mathrm{for}}\uparrow$|$Acc_{\mathrm{mem}}\uparrow$|
> > |BBT|$79.38_{\pm 0.01}$|$87.30_{\pm 0.01}$|$71.09_{\pm 0.00}$|$58.75_{\pm 0.01}$|$88.98_{\pm 0.04}$|$43.85_{\pm 0.01}$|$94.22_{\pm 0.05}$|$90.17_{\pm 0.08}$|$99.06_{\pm 0.01}$|
> > |BBT w/ Sep-CMA-ES|$69.29_{\pm 0.01}$|$67.83_{\pm 0.02}$|$70.81_{\pm 0.01}$|$53.74_{\pm 0.02}$|$74.72_{\pm 0.02}$|$41.96_{\pm 0.02}$|$91.18_{\pm 0.02}$|$84.44_{\pm 0.04}$|$\\textbf{99.07}_{\pm 0.00}$|
> > |BBT w/ VkD-CMA-ES|$75.41_{\pm 0.02}$|$79.56_{\pm 0.01}$|$\\textbf{71.67}_{\pm 0.01}$|$55.12_{0.02}$|$81.49_{\pm 0.03}$|$41.65_{\pm 0.02}$|$91.25_{\pm 0.05}$|$84.58_{\pm 0.09}$|$99.06_{\pm 0.00}$|
> > |Ours|$\\textbf{80.99}_{\pm 0.01}$|$\\textbf{93.37}_{\pm 0.02}$|$71.52_{\pm 0.01}$|$\\textbf{59.67}_{\pm 0.01}$|$\\textbf{89.29}_{\pm 0.01}$|$\\textbf{44.81}_{\pm 0.01}$|$\\textbf{97.28}_{\pm 0.01}$|$\\textbf{95.94}_{\pm 0.01}$|$98.67_{\pm 0.01}$|
> >
> > As we can see, the variants of BBT do not provide stable performance over different datasets. In particular, the three BBT variants show $Err_\text{for}$ more than 5-10% lower than our method on ImageNet30. Furthermore, the results on CUB-200-2011 highlight the difficulty of the Black-Box Forgetting problem, as even our method yielding the best results could achieve only 59.67% in $H$.
> >
> > Moreover, as shown in Tables 1 and G, all the existing methods failed to achieve 80% in $Err_\text{for}$ for all the datasets. The performance degradation in $H$ is also significant, with $H$ values for BBT and CBBT being up to about 10% lower than for Ours. Figure 3 suggests that the scalability of BBT is poor, which is an inherent limitation that can be resolved by our method.
> >
> > Based on these facts, we believe that our Black-Box Forgetting is not an insignificant problem, but one that is both novel and worthy of technical consideration.
> >
> > [a] Kang et al., Few-shot Object Detection via Feature Reweighting, in Proc. ICCV, 2019.
> >
> > [b] You et al., Universal Domain Adaptation, in Proc. CVPR, 2019.
> >
> > [c] Yu et al., Multi-Task Curriculum Framework for Open-Set Semi-Supervised Learning, in Proc. ECCV, 2020.

---

> ### Comment · Reviewer_SKod · 2024-08-14
>
> The authors' statement on the novelty of problem setup is somewhat convincing, and I want to express my huge gratitude for the extensive responses. However, I would regretfully say that I want to keep my score.
>
> This black-box forgetting problem has not been explored yet, and the authors address it with their new method.
> The proposed method, latent context sharing (LCS), is designed to reduce the number of learnable parameters for better black-box optimization.
> I still do not see the necessity of this proposal, and there is a missing link in the connectivity between LCS and the black-box forgetting problem.
>
> To be specific, the LCS method is not tailored for "forgetting," and it can be regarded as a method for improving parameter efficiency that will be applied to other PEFT scenarios.
> The weak connection between the LCS method and the problem setup is the main reason that I kept my score towards rejection.

---

> > ### Author Response · Authors · 2024-08-14
> > **Response to Reviewer SKod's comment**
> >
> > Thank you for your response. We are very happy that now Reviewer SKod acknowledged the novelty of the Black-Box Forgetting problem.
> >
> > Let us clear up the reviewer's misunderstanding: LCS is designed for forgetting. Please see Figure 5 in our original paper. The results show that as the ratio of the number of classes to be forgotten $r_\text{for}$ increases, the forgetting performance $Err_\text{for}$ of the existing parametrization method, BBT, tends to decrease. The only way to improve the $Err_\text{for}$ of BBT is to increase the number of learnable parameters $m$ (as can be seen in Figure 3), which clearly suggests the limitation of the scalability of BBT. Based on this observation, we developed our LCS to fundamentally improve the scalability of parameter representation for improving $Err_\text{for}$, that is, for forgetting.
> >
> > Note that the reviewer's comment is never a disadvantage of our LCS, even though it may suggest that LCS can be applied to other problems, besides forgetting.

---

> ### Comment · Reviewer_SKod · 2024-08-14
>
> I am so sorry for my decision-invariance attitude, but I want to adhere to my rating.
>
> These long discussions might cause your mind severe damage, and I am regretful for keeping my statement due to the unsolved concern.
>
> I can not agree with the authors' claim that "LCS is designed for forgetting." Figure 5 is just an empirical result, where the authors provide post-experiment interpretation based on the results.
>
> To be specific, it is only a consequential interpretation, and the design of the methodology cannot be justified as it is for forgetting. That is why I think the connection between the problem definition and the derived methodology is weak, and I will try to stick to this position.
>
> However, since I found some good implications from this study and other reviewers are looking positively, I will not be upset even if this paper is accepted.
>
> Sorry again,
>
> Reviewer SKod

---

> > ### Author Response · Authors · 2024-08-14
> >
> > We would like to thank Reviewer SKod for your patience and your consideration during this rebuttal period. But could you please allow us to provide one last response, because we feel that the reviewers' view is somewhat biased, saying like "good methods should not be inspired by empirical observations."
> >
> >
> > Looking at the facts of the past, there are a number of good methods built on empirical observations (and are not designed specific to the target tasks). For example, Focal Loss was originally proposed for object detection, but it is actually a loss to mitigate class imbalance, and its connection to object detection is rather weak [a]. ArcFace [b] was proposed for face verification, but its core is a variant of max margin loss, which is not really relevant to faces.
> >
> >
> > Our method is designed based on sharp observations in forgetting, which in itself should not be punished. Furthermore, even if the reviewer could not agree that the LCS is designed for forgetting, at least could agree that it is designed for the Black-Box setting, which is the other half of the focus of this paper.
> >
> >
> > [a] Lin et al., Focal loss for dense object detection, in Proc. ICCV, 2017.
> >
> > [b] Deng et al., ArcFace: Additive Angular Margin Loss for Deep Face Recognition, in Proc. CVPR, 2019

---

### Official Review · Reviewer_bJB5 · 2024-07-11

**Soundness:** 2
**Presentation:** 2
**Contribution:** 2
**Rating:** 6
**Confidence:** 3

**Summary:**

This paper studies the black-box forgetting problem. To this end, the authors optimize the input prompts with a proposed latent context sharing scheme by CMA-ES optimization. The authors achieve the selective forgetting goal by minimizing the cross-entropy loss on memorized classes and maximizing entropy loss on forgetting classes. Experiments demonstrate the promising of the proposed method.

**Strengths:**

The model forgetting/unlearning problem for black-box models is both interesting and practical. (However, the specific forgetting setting in this work is still unclear to me.)

Compared with BBT [Sun et al.], the proposed Latent Context Sharing (LCS) scheme effectively enhances the learning ability of CMA-ES, providing a promising technique for CMA-ES-based black-box model fine-tuning.

Sensitivity analyses regarding hyper-parameters are sufficient.

**Weaknesses:**

The problem setting of this work does not convince me, possibly due to the presentation of the paper. I hope the authors can provide more justification regarding this.

Experiments could be further enhanced, as my detailed comments below.

**Questions:**

Q1.

*“—Retaining the classes that do not need to be recognized may decrease overall classification accuracy, as well as cause operational disadvantages such as the waste of computational resources and the risk of information leakage”*

How do you define the overall accuracy? Is it the overall accuracy of all classes of the pre-trained model, or the overall accuracy of downstream tasks, such as CIFAR-10 in the experiments? This is quite confusing for me. If it is the latter, during the fine-tuning process on downstream tasks, it seems that tuning only the necessary classes (those to be recognized) can achieve the best performance. Why must we forget some classes?

Moreover, the phrase ‘waste of computational resources’ is also confusing. Learning on all classes is conducted during the pre-training phase and cannot be modified by the proposed method. Then, during fine-tuning, why would forgetting to reduce computational resources? **Aside from preventing information leakage**, what is the rationale behind forgetting certain classes?  i.e., When we fine-tune a CLIP model on CIFAR-10, what is the real and sound motivation for us to forget some classes of CIFAR-10?

Q2. If the final goal is to forget 40% of classes and memorize 60% of classes on CIFAR-10, why must we fine-tune CLIP? Why not directly train a new model (with the same architecture as CLIP or a smaller ResNet or ViT) with only the 60% memorized classes? It would be better to include the results of this baseline in Table 1. I assume that the authors believe a model fine-tuned from CLIP could achieve better performance on the memorized classes than a model trained from scratch. If this is the case, experiments on CIFAR datasets alone are insufficient to demonstrate this, as these data are relatively easy. Experiments on more complex datasets/tasks, such as those mentioned in autonomous driving, are necessary to further help to justify the superiority of the proposed method.

Q3. Can the proposed method be extended to more complex tasks such as segmentation or object detection? As the authors mentioned, in an autonomous driving system, it is sufficient to recognize a limited number of classes such as cars, pedestrians, and traffic signs.  In autonomous driving, the task is often more complex than pure classification, such as 2D and 3D object detection tasks.

Q4. How about the performance of Ours vs. Solely Fine-tune CLIP on memorized classes? It would be much better if this result is verified on more complex datasets.

Q5. In the white-box setting, one can update more model parameters beyond just the input prompts. How would the performance change if we update all model parameters for CoOp (White-Box)?


I would like to increase my score if the authors could convince me regarding the problem setting.

**Limitations:**

The proposed method is not fully black-box, as it still requires access to the embeddings and output logits, which are unavailable in a fully black-box setting, similar to the GPT-4 API. However, I understand that achieving a fully black-box setting is very challenging, and I do not consider this a reason for rejection. I mention this point merely as a slight limitation.

---

> ### Author Rebuttal · Authors · 2024-08-06
>
> ### Q1. Why must we forget some classes?
> We assume that the reviewer already acknowledged the benefit of preventing information leakage through forgetting. In addition to this, we here would like to emphasize the potential benefits of exploring selective forgetting.
> 1. Toward addressing the "Right to be Forgotten": If a service provider is asked to remove information so that their model cannot recognize certain information, it might need to comply with the request. This can be accomplished by training the model from scratch by removing samples of that class from the training data. However, retraining a large-scale model consumes an enormous amount of energy, which should be avoided. Selective forgetting may provide an efficient solution to this problem.
> 2. Toward efficient large-scale pre-trained models: Improving the efficiency of large-scale pre-trained models is of current interest to many researchers. Various attempts have been made such as model compression and architecture optimization (e.g., https://sites.google.com/view/elvm/program). Meanwhile, as the "scaling law" indicates, the reasonable size of a model correlates with the amount of knowledge it memorizes. Therefore, if the number of classes (amount of knowledge) that can be recognized by the model is limited, the model can be scaled down accordingly, thereby improving the efficiency of the model. This contributes to expanding the applicability of large-scale pre-trained models.
> 3. Toward better control over text-to-image generation: While diffusion-based text-to-image generation can generate diverse types of high-quality images, controlling the content of images remains a challenge. Recent research has focused on "forgetting" visual concepts in order to avoid generating undesirable content [a-c]. These methods forget certain classes by directly fine-tuning the diffusion model, but tuning the diffusion model itself is often costly. In contrast, given that typical generative models use a text encoder of a pre-trained VLM (e.g., CLIP) for conditioning, our method may provide an efficient approach to class forgetting by fine-tuning only the prompts of the text encoder.
>
> We believe that our method will open new directions for these important problems of interest to the ML community, even if these are not immediately feasible with this paper alone. We would like to add the discussion as a broad impact of this work in the final version.
>
> [a] Heng et al., Selective Amnesia: A Continual Learning Approach to Forgetting in Deep Generative Models, In Proc. NeurIPS, 2023.
>
> [b] Lu et al., Mace: Mass Concept Erasure in Diffusion Models, In Proc. CVPR, 2024.
>
> [c] Zhang et al., Forget-me-not: Learning to Forget in Text-to-Image Diffusion Models, In Proc. CVPR, 2024.
>
> ### Q2. Forgetting vs. Learning from scratch.
> As requested by the reviewer, we show the accuracy of ResNet-18 and ViT-B/16 trained from scratch over only the classes to be memorized in Table D in the attached PDF. As can be seen, both models cause severe overfitting to the training data. That is, while these models achieve reasonable accuracy on the training data, they exhibit severely poor performance on the test data. We also tested both models when initialized with ImageNet pretrained weights. While the results improve somewhat for ResNet-18, these are still far behind our forgetting-based method. The reason for this is that, following the common protocol in context optimization [a], we conducted our experiments in few-shot scenarios as explained in Line 201, which overwhelmingly lacks the number of training samples to learn the weights for even ResNet-18 and Vit-B/16.
>
> [a] Zhou et al., Learning to Prompt for Vision-Language Models, IJCV, 2022.
>
> ### Q3. Is the method applicable to detection / segmentation?
> It would be possible by combining with some detectors / segmentors. For example, recent approaches in object detection rely on a two-stage framework [a,b], which first uses region proposal network to get object proposals and then applies zero-shot CLIP to each proposal to identify the object class. Replacing the zero-shot CLIP with one tuned by our method would prevent detecting classes that have been forgotten.
>
> [a] Zhao, et al. Exploiting unlabeled data with vision and language models for object detection, in Proc. ECCV, 2022.
> [b] Zhao, et al. Taming Self-Training for Open-Vocabulary Object Detection, in Proc. CVPR. 2024.
>
> ### Q4. Ours vs. Solely Fine-tune CLIP on memorized classes?
> We tested the performance of Solely Fine-tune CLIP on memorized classes on four datasets including ImageNet30. The results are shown in Table E in the attached PDF. As we can see, ours and Solely Fine-tune CLIP are comparable in $ACC_\text{mem}$ for the three datasets, except for CUB-200-2011. That is, even if we perform forgetting, it does not significantly impair accuracy on the memorized classes.
>
> ### Q5. In the white-box setting, one can update more model parameters beyond just the input prompts.
> Yes, but it does not work. We tested "CoOp (White-box) + Parameter Update", i.e., updating not only the learnable contexts in the prompt but also the model parameters. The results are shown in Table F in the attached PDF. We see that simultaneously updating the model parameters does not improve performance, but rather hurts it. This is not surprising, as it is known that straightforward fine-tuning of the zero-shot CLIP does not improve performance [a].
>
> [a] Wortsman et al., Robust Fine-Tuning of Zero-Shot Models, in Proc. CVPR, 2022.
>
> ### Q6. The proposed method still requires access to the embeddings and output logits, I do not consider this a reason for rejection.
> Thank you for this fair consideration of the limitation described in our original paper (Sec. 6). In fact, existing black-box tuning methods such as BBT and CBBT assume the same setup, which we followed. On the other hand, the consideration of a complete black-boxing approach is an interesting challenge that we would like to pursue.

---

> > ### Comment · Reviewer_bJB5 · 2024-08-12
> > **Response to authors**
> >
> > Thanks for the authors’ thoughtful response. With newly provided results, the paper becomes much stronger on the empirical side. The clarifications regarding the significance of Selective Forgetting on Generative Models make the problem setting more practical. Most of my initial concerns have been addressed. I shall increase my score and please include the new results and discussions/clarifications in the revised paper.
> >
> > Moreover, for results in Table E, when the pre-trained model’s performance is not good enough, solely fine-tuning on memorized classes still achieves much better Acc_{mem} than the proposed method, i.e., results on CUB-200-2011. This could be further discussed in the revision.

---

> > > ### Author Response · Authors · 2024-08-12
> > > **Thank you, and our response to your additional question on Table E.**
> > >
> > > We would like to thank you for your great effort in carefully reading our rebuttal and for your sincere consideration of it. We are very happy that our response has successfully resolved your questions. Of course, we will make sure to include our new experimental results, discussions, and explanations we provided in our rebuttal in the final version.
> > >
> > > Regarding your additional question about the results in Table E, our response to **Q6 of Reviewer VErt** should be highly relevant. That is, Table C in the attached PDF shows that $Err_\text{for}$ and $Acc_\text{mem}$ are in a trade-off relationship, i.e., Ours ($Acc$ prio.), which is optimized by prioritizing $Acc_\text{mem}$, successfully improves $Acc_\text{mem}$, but at the expense of $Err_\text{for}$. This suggests that the underlying reason why the pre-trained model in Table E achieved higher $Acc_\text{mem}$ than Ours is probably because Ours was forced to sacrifice $Acc_\text{mem}$ to increase $Err_\text{for}$.
> > > Nevertheless, our method achieves a better trade-off (i.e., higher $H$) than the existing methods (BBT and CBBT) in all of the experiments. Furthermore, despite being a black-box method, Ours ($Acc$ Prio.) achieves $Acc_\text{mem}$ comparable to CoOp, a white-box method, which emphasizes the effectiveness of our method as a black-box optimization method. We will add a clear discussion about this point in the final version as well. Thank you for your valuable suggestion again!

---

### Official Review · Reviewer_VErt · 2024-07-15

**Soundness:** 3
**Presentation:** 3
**Contribution:** 3
**Rating:** 6
**Confidence:** 4

**Summary:**

This paper addresses the problem of selective forgetting of specified classes, which involves tuning a pre-trained model to reduce the classification accuracy for only the specified classes without affecting the accuracy for the others. The proposed model introduces a novel method for Black-Box Forgetting based on the derivative-free optimization of a learnable text prompt. It introduces Latent Context Sharing (LCS), a novel parameterization method for contexts, which mitigates the difficulty of high-dimensional optimization using derivative-free optimization.

**Strengths:**

1- The proposed method explores the selective forgetting problem for CLIP-based models (PTMs), where the task is to make the model unable to recognize only the specified classes while maintaining accuracy for the others. More importantly, it addresses a novel problem of selective forgetting for black-box models, termed Black-Box Forgetting, and proposes an approach to solve this problem.

2- It proposes Latent Context Sharing, which introduces common low-dimensional latent components among multiple tokens for the prompt to reduce complexity.

3- Instead of optimizing the model parameters, it proposes a novel method for Black-Box Forgetting based on derivative-free optimization of a learnable text prompt, which avoids the need for information about the pretrained model.

**Weaknesses:**

1- The proposed approach for forgetting selective classes from the pre-trained model requires data samples for each elective class. Since it is applied in CLIP-based multi-modals, where the text encoder is also available, one might wonder why data samples are necessary for each forgetting class. Could the same result not be achieved using just the class name?

2- The proposed approach claims that it introduces Latent Context Sharing (LCS), a novel parameterization method for contexts, to mitigate the difficulty of high-dimensional optimization. However, considering that the embedding dimension in CLIP-based models is 512, which is not excessively large, and that this approach reduces the dimension to 100 for the CUB dataset, the reduction does not seem significantly lower than the original CLIP dimension. Therefore, in the case of larger datasets, this reduction might not have a substantial impact compared to the original CLIP dimension.

3- The proposed approach claims to be a black box due to the unavailability of pre-trained model information, such as architecture, parameters, and gradients, during training. However, my concern is that if you are given the pretrained model, extracting information about the model architecture and its parameters can still be quite challenging?

4- How is the proposed black-box forgetting different from machine unlearning?

5- I recommend conducting experiments on more diverse datasets, such as ImageNet-1K, to gain a better understanding of this approach.

6- Why is Acc_mem lower than the baseline approaches for each dataset?

7- Can the proposed approach be applied in a continual learning setup?

**Questions:**

Please address all my concerns raised in the weaknesses section.

**Limitations:**

My concerns raised in the weaknesses section described the limitations of the proposed approach.

---

> ### Author Rebuttal · Authors · 2024-08-06
>
> ### Q1. Could the same result not be achieved using just the class name?
> Good suggestion! We tried to tune the latent contexts by only using the class names (i.e., class embeddings). Specifically, let $z_c$ and $z$ denote the class embeddings before and after prompt tuning for the class to be forgotten, respectively. Only $z$ is trainable. We aim to tune $z$ by minimizing the following negative NT-Xent loss, $ \log \frac{\exp(z_c ^\top z / \tau)}{\sum_i \exp(z_i ^\top z / \tau)}$, where $z_i$ is the class embedding for the $i$-th class to be kept memorized. This loss requires $z$ to be orthogonal to $z_c$ as well as be similar to the embeddings of the other classes $\\{z_i\\}$. The results are reported in Table B in the attached PDF. We found that Ours is much better than the approach using only the class embeddings (C-Emb.), which proves that tuning with only the class embeddings does not provide satisfactory performance.
>
> ### Q2. 512-D should not be high-dimensional.
> Derivative-free optimization based on multi-point search, such as CMA-ES, is directly affected by the curse of dimensionality, so the number of dimensions exceeding, typically 10-D, is often considered high-dimensional [a]. This is why the existing black-box tuning method, BBT, optimizes low-dimensional (around 10-D) latent contexts instead of the raw 512-D embeddings.
>
> [a] Hansen et al., Reducing the Time Complexity of the Derandomized Evolution Strategy with Covariance Matrix Adaptation (CMA-ES). Evolutionary Computation, Vol. 11, No. 1, pp. 1-18, 2003.
>
> ### Q3. Given the pretrained model, extracting information of the model should not be challenging.
> Following the same setup as in previous black-box tuning (e.g., BBT and CBBT), we assume that no pre-trained model is given and can only access to the input/output of the pre-trained model (e.g., via an API). In such a case, we cannot know the internal information of the model and cannot extract information about the model architecture or its parameters.
>
> ### Q4. Machine Unlearning vs. Black-Box Forgetting.
> These two are closely related but different. Machine Unlearning typically aims to remove the influence of specified training samples on the training model, whereas Black-Box Forgetting aims to prevent the recognition of specified classes. Furthermore, we in this paper address the black-box setting, which has not yet been well-traveled in machine unlearning.
>
> ### Q5. Experiments on more diverse datasets, such as ImageNet-1K?
> Although we were not able to complete the experiment with ImageNet-1K in this short rebuttal period, we instead report the results of additional experiments with its subset, ImageNet30 [a]. The results shown in Table G in the attached PDF demonstrate that
> our method outperforms all the methods in $H$. We will add the results on ImageNet-1K in the final version.
>
> [a] Hendrycks et al., Using Self-Supervised Learning Can Improve Model Robustness and Uncertainty, in Proc. NeurIPS, 2019.
>
> ### Q6. Why is $Acc_\text{mem}$ lower than the baselines?
> This is because $Err_\text{for}$ and $Acc_\text{mem}$ are in a trade-off relationship, with $Acc_\text{mem}$ tending to decrease as $Err_\text{for}$ is increased. This is presumably because features between classes are not completely disentangled in the feature space, so forgetting one class may negatively affect other classes (just as dog and cat share come common features). To provide justification for this, we report the results of using a loss more prioritized (weighted) for $Acc_\text{mem}$ in our method in Table C as "Ours ($Acc$ prio.)" in the attached PDF. We can see that Ours ($Acc$ prio.) outperforms all the other methods in $Acc_\text{mem}$, with sacrificing $Err_\text{for}$. Notably, both Ours and Ours ($Acc$ prio.) outperform BBT and CBBT in $H$, indicating that our method achieves a better trade-off than BBT and CBBT.
>
> ### Q7. Can the proposed approach be applied in a continual learning setup?
> Thank you for your interesting suggestion! While application of our method to continual learning is not in the scope of this paper, theoretically we think it is possible. Our method is based on a context optimization as CoOp. So it can readily be coupled with some of standard continual learning methods, such as distillation-based [a].
>
> [a] Li et al., Learning without Forgetting, IEEE TPAMI, Vol., 40, No.12, pp. 2935-2947, 2017.

---

### Official Review · Reviewer_LjLN · 2024-07-15

**Soundness:** 3
**Presentation:** 2
**Contribution:** 3
**Rating:** 6
**Confidence:** 4

**Summary:**

The authors propose to apply selective forgetting to black-box pretrained models, instead of the usual white-box settings. Since it is a black-box method, there is no parameter update and instead the prompts are the ones being optimized to decrease the performance on the target class to be forgotten. This is done by using an existing covariance matrix adaptation, which is a derivative-free optimizer. The paper describes the issues of optimizing a large dimensional space while gradients are not available, and the problem of having samples from the target forgotten class being mostly pushed towards a close-by, non-forgotten class instead. The first issue is solved by splitting the latent context between unique and shared, and reparametrizing them individually to lower dimensions. The second issue is solved by excluding the label information from a classic CE loss, and adding it as regularization for maximizing the entropy of the overall confidence.

**Strengths:**

Selective forgetting and machine unlearning are of interest within our community, and thus this paper fits very well with the conference. The need to move from white-box to black-box methods is something natural, as it has happened before with topics like adversarial attacks, so it is relevant to explore how to tackle this more complex setting. In terms of originality, previous work is well referenced to establish the originality of the paper comes from the proposed strategy adapting to the needs of the proposed novel scenario. The introduction of the scenario and its needs, as well as most of the method explanations are well described and are easy to follow. The limitations are very well covered, and the significance of the paper is well motivated and clear.

**Weaknesses:**

Despite the main idea being clear and most of the method being well described, the writing could use some revision. CMA-ES is quite a key component of the proposed strategy, but is covered quite shortly. The LCS (ii) part is a bit over-complicated for such an easy concept. I think it could be explained a bit more elegantly and would highlight the usefulness and easiness of having this type of parametrization. Same goes for some of the subsections within the Analysis (4.3). I assume that the lack of space was the reason why some of the concepts from the compared methods are just mentioned and not explained in detail. However, in order to make it easier for the reader to understand what the analysis refers to and drive some of the conclusions, it might be better to just move one or two of them to the appendix. My suggestion would be moving 4.3.1 and 4.3.5 to the appendix, and using that space to improve the analysis of the others.

Nomenclature in some cases is a bit confusing. In figure 1, the losses are called forget and memorize, while in equations (1) and (2*missing) they are called uniform and CE. Same with z having subscripts or superscripts depending on BBT or LCS.

Text in figures 3 and 4 is too small.

**Questions:**

1. In the introduction, it is shortly mentioned, but I would like some more context on how close are some of the machine unlearning settings to the one proposed in this submission.

2. Footnote 2 is a bit weak. How are the results shown in section 4 enough reason to assume that there is a composition of ULC and SLC?

3. Why are the latent contexts optimized for the hyperparameters used? Are those decided under a validation set or overfitted to the test? Same for the number of latent contexts.

4. In appendix A.2, what is the reasoning for the groupings of classes to be forgotten? Just random? Or do they relate in some way that can support the use of equation (2*missing) versus (1)?

**Limitations:**

Regarding the experimental statistical significance, the authors mention that they do not report error bars, and I would also consider that such set of experimental settings would require the need to run each method multiple times, specially when there are some results that are quite comparable and could lead to statistical significance showing that some methods is not significantly different than another under some metrics.

This is such a strong point, in my opinion, that it is the reason why I rate the paper down from a 7 (accept) to a 6 (weak accept). Given that the paper relies quite heavily on its experimental analysis to showcase the interesting proposed scenario compared to current zero-shot or white-box approaches. Also in the appendix, there is this random sampling, which does not report how many times it is sampled either. Finlly, the point from question 2 is also important to showcase the limitation on the argumentation of the existence of ULC and SLC.

---

> ### Author Rebuttal · Authors · 2024-08-06
>
> ### Q1. CMA-ES is covered quite shortly.
> We will expand the description of CMA-ES in the first paragraph of Line 105 in Sec. 3 as follows:
>
> > We employ CMA-ES, a widely used evolutionary algorithm for black-box optimization in continuous, because a textual prompt to be optimized is a continuous variable. CMA-ES is a multi-point search algorithm based on a multivariate normal distribution and proceeds the search by iterating (i) sampling candidate solutions, (ii) evaluating the loss values of the candidates, (iii) weighting the candidates based on the loss values, and (iv) updating the mean and covariance matrix of the distribution by using the weighted candidates. Due to the nature of multi-point search, the performance of CMA-ES degrades in high-dimensional problems, typically ten or more dimensions [a,b]. While several extensions have been proposed, e.g., [a,b], these methods require knowledge of independence among variables, which is not always known. In this paper, we propose a customized extension of CMA-ES to Black-Box Forgetting.
>
> [a] Ros and Hansen, A Simple Modification in CMA-ES Achieving Linear Time and Space Complexity, in Proc. PPSN, 2008.
>
> [b] Akimoto and Hansen, Projection-based Restricted Covariance Matrix Adaptation for High Dimension. in Proc. GECCO, 2016.
>
> ### Q2. LCS could be explained a bit more elegantly.
> To facilitate intuitive understanding of our LCS, we add at the beginning of the LCS section (Line 139) the inspiration behind our LCS and the justification for the composition of ULC and SLC.
> > Fig. 2c shows the overview of LCS. The key idea is to assume shared parameters among different latent contexts. This inspiration comes from successful word embedding methods; most word embedding methods are trained on the assumption that locally co-occurring words have semantic correlations between them (e.g., [a-c]). This inspires the idea of explicitly modeling semantic correlations between words in a prompt as shared components.
>
> [a] Mikolov, Efficient Estimation of Word Representations in Vector Space, arXiv pre-print 1301.3781, 2013.
>
> [b] Pennington, et al., GloVe: Global Vectors for Word Representation, in Proc. EMNLP, 2014.
>
> [c] Devlin et al., BERT: Pre-training of Deep Bidirectional Transformers for Language Understanding, in Proc. NAACL-HLT, 2019.
>
> ### Q3. Sec. 4.3 could be reorganized.
>
> As suggested, we will move 4.3.1 and 4.3.5 to the appendix to put priority to the analyses on the effectiveness of the parametrization of our LCS. Thanks!
>
> ### Q4. Some nomenclature and texts in figures and equations could be improved.
> As suggested, we will revise the numbering of Eq. 2 and the name of the loss (i.e., change "CE" in Fig. 1 to "uniform") in the final version. The text in Fig. 1 will be enlarged. The superscript of $z$ is to distinguish shared components from unique components, so is essential for our LCS.
>
> ### Q5. Some more contexts on machine unlearning?
> We will add a review of machine unlearning in the Related Work section in the final version:
> > Machine unlearning aims to remove an arbitrary sample from a pre-trained model, i.e., obtaining a model that is identical to the one trained from scratch without that sample [a-g]. Many methods have been proposed, for example, to construct a forgettable model by transforming the learning algorithm into a sum of the training samples [a], to achieve forgetting by linear approximation of a nonlinear model [b], and to update the model to be closer to / farther from the original model in the retain / forget samples [e]. Methods specific to certain learning algorithms such as LDA [f] and SVM [g] have also been explored. Machine unlearning and Black-Box Forgetting are closely related but different; Machine unlearning aims to remove the influence of specified training samples on the training model, whereas Black-Box Forgetting aims to prevent the recognition of specified classes. Forgetting specified classes has attracted much attention recently in various contexts [h-l]. We in this paper address the black-box setting, which has not yet been explored.
>
> [a] Cao and Yang, Towards Making Systems Forget with Machine Unlearning. In Proc.IEEE Symp. Security and Privacy, 2015.
>
> [b] Golatkar et al., Mixed-Privacy Forgetting in Deep Networks, In Proc. CVPR, 2021.
>
> [c] Sekhari et al., Remember What You Want to Forget: Algorithms for Machine Unlearning, In Proc. NeurIPS, 2021.
>
> [d] Bourtoule et al., Machine Unlearning, In Proc. IEEE Symp. Security and Privacy, 2021.
>
> [e] Kurmanji et al., Towards Unbounded Machine Unlearning, In Proc. NeurIPS, 2024.
>
> [f] Guo et al., Certified Data Removal from Machine Learning Models, In Proc. ICML, 2020.
>
> [g] Chen et al., A Novel Online Incremental and Decremental Learning Algorithm based on Variable Support Vector Machine, Cluster Computing, 2019.
>
> [h] Heng et al., Selective Amnesia: A Continual Learning Approach to Forgetting in Deep Generative Models, In Proc. NeurIPS, 2023.
>
> [i] Lu et al., Mace: Mass Concept Erasure in Diffusion Models, In Proc. CVPR, 2024.
>
> [j] Zhang et al., Forget-me-not: Learning to Forget in Text-to-Image Diffusion Models, In Proc. CVPR, 2024.
>
> [k] Shibata et al., Learning with Selective Forgetting, in Proc. IJCAI, 2021.
>
> [l] Ye et al., Learning with Recoverable Forgetting, in Proc. ECCV, 2022.
>
> ### Q6: Hyperparameters tuned on validation sets or test sets?
> All the hyperparameters were tuned on validation sets. We will clarify this in the final version.
>
> ### Q7: Classes to be forgotten determined at random or intentionally?
> These were determined randomly for fairness. We will clarify this in the final version.
>
> ### Q8: No error bars reported.
> Good suggestion! All the results in our paper are averages of three runs with different random seeds. Check out Table A in the attached PDF, which is Table 1 of our submission with the standard deviations added. Ours clearly outperforms the compared methods. We will add standard deviations to all the results in the final version.

---

> > ### Comment · Reviewer_LjLN · 2024-08-12
> >
> > Thanks for the author's response. I agree that answers to Q1-Q5 will help improve the manuscript. Specialy, the context on machine unlearning, and the differences from the proposed black-box forgetting scenario are relevant, and the proposed changes on literature review are well covered. However, reading through the other reviewer's and response, I would stress the need to highlight well those differences.
> >
> > (a) As further discussion, when saying "_Machine unlearning aims to remove the influence of specified training samples on the training model, whereas Black-Box Forgetting aims to prevent the recognition of specified classes._", it raises the question of how that affects the evaluation. If the shift between the two problems is going from specific samples to whole classes, how well do the metrics proposed evaluate the new scenario? The metrics used on the paper are based on the accuracy over specific samples, instead of measuring the shift between using the learned latent contexts and not. I understand that the metrics on single samples are aggregated, I am referring towards having a better metric to measure that the distribution of the specified class is forgotten.
> >
> > (b) Q6, Q7 and Q8 are satisfactory responses for the experimental setting and evaluation. I would support adding those details to the final version, since the scenario should be clearly stated for further reproducibility. I appreciate that the results are over more than one seed, and I think the low standard deviation is a positive sign for the conclusions of the experimental section and reinforce the conclusions.
> >
> > (c) My original question 2 is unaddressed. The paper states "_We assume that each latent context is composed of unique components (Unique Latent Contexts (ULC)) and common components (Shared Latent Context (SLC)) among multiple latent contexts_" with a footnote saying that the assumption is true due to the experiments in Sec. 4. However, this seems like a very weak argument. How did this assumption came to be, or what supports it, even before having an experimental support for it?
> >
> > (d) Reading through some reviews and rebuttals, the author's have also clarified some of the interesting questions raised by fellow reviewers, which in most cases are satisfying. Answer to Q1 of reviewer VErt is not very convincing. If the proposed scenario distinguishes itself from machine unlearning by moving away from specific samples, the idea of using the prompts becomes much more interesting, instead of keeping samples. The authors address on their response the performance improvements, but the interesting part of the question is not the numbers, but challenging the decision of using the given samples vs. sampling the prompts for the classes involved.

---

> > > ### Author Response · Authors · 2024-08-13
> > > **Our responses to your additional questions (b) and (c).**
> > >
> > > Thank you for your detailed review not only of our responses, but also of the other reviewers' comments and our responses to them. We are very pleased to hear that most of our responses were satisfactory. Below we would like to respond to your additional questions, first to (b) and (c), followed by (a) and (d).
> > >
> > > **To (b)**: Yes, we will make sure to add the results and discussions, as well as the detailed evaluation protocol in the final version.
> > >
> > > **To (c)**: Sorry for the lack of a detailed explanation. The following statement, which is originally provided as our response to question 2, is the inspiration behind our design. Please let us know if it is still not convincing to you.
> > >
> > > > The key idea is to assume shared parameters among different latent contexts. This inspiration comes from successful word embedding methods; most word embedding methods are trained on the assumption that locally co-occurring words have semantic correlations between them (e.g., [a-c]). This inspires the idea of explicitly modeling semantic correlations between words in a prompt as shared components.
> > >
> > > [a] Mikolov et al., Efficient Estimation of Word Representations in Vector Space, arXiv pre-print 1301.3781, 2013.
> > >
> > > [b] Pennington et al., GloVe: Global Vectors for Word Representation, in Proc. EMNLP, 2014.
> > >
> > > [c] Devlin et al., BERT: Pre-training of Deep Bidirectional Transformers for Language Understanding, in Proc. NAACL-HLT, 2019.

---

> > > > ### Author Response · Authors · 2024-08-13
> > > > **Our responses to your additional questions (a) and (d).**
> > > >
> > > > **To (a) and (d):** We appreciate you sharing these interesting questions with us.
> > > >
> > > > We assume that your intention behind these two questions is that "the proposed method performs prompt tuning with a few training samples, but to truly realizing class-selective forgetting, it would be necessary to control the class distribution, rather than relying on training samples." Given that the fundamental goal of machine learning is to obtain generalization performance from a limited number of training samples, it would be rather difficult to cover this question in this paper alone, but we will try to discuss as much as possible.
> > > >
> > > > First of all, would it be possible to tune a pre-trained VLM without using training samples? As is well known, one advantage of tuning the prompt rather than the model parameters is that generalization performance can successfully be improved even when only a few training samples are available [a]. However, prompt tuning without using training samples, i.e., "zero-shot prompt tuning," is still an open problem; effective tuning is difficult unless some external knowledge is available, e.g., LLMs [b] or style-specific prompts [c].
> > > >
> > > > In fact, we tried to answer Q1 of VErt by updating the class embeddings by minimizing the negative NT-Xent loss, without using training samples. Indeed, the reason for using the negative NT-Xent loss was exactly what you intended, i.e., we aimed at controlling the distribution of the class embeddings without relying on any training samples. More specifically, to ensure that the embedding of the class to be forgotten is placed equidistant from the embeddings of all the other classes. We believe this is a reasonable idea, provided that no external knowledge is available.
> > > >
> > > > However, as the results show in Table B, $Acc_\text{mem}$ was much lower than when using the training samples. This is probably due to the nature of the negative NT-Xent loss, whereby the embedding of the class to be forgotten is closer to the embeddings of the other classes to be memorized, resulting in more misclassifications.
> > > >
> > > > Given these observations, it would be challenging to fully control the class distributions without any training samples. This also implies that developing new evaluation metrics that do not rely on test samples is rather challenging for now, although it is an interesting problem. However, since classification performance is usually evaluated on test samples, we believe that the evaluation metrics we used in our paper are at least reasonable.
> > > >
> > > > Next, how should we decide whether to use given training samples or to directly control the class embeddings? The advantage of controlling class embeddings is that forgetting can be achieved even for the classes without training samples. Therefore, a natural recommendation comes up in our mind is to use training samples for the classes for which those are available, and to directly control the class embeddings for the classes with no training samples.
> > > >
> > > > To test the validity of this recommendation, we experimented under conditions in which half of the classes to be forgetting had training samples available and the rest did not. The results on ImageNet30 are shown below.
> > > >
> > > > | Method                   |    $H\uparrow$    | $Err_{\mathrm{for}}\uparrow$ | $Acc_{\mathrm{mem}}\uparrow$ |
> > > > | :----------------------- | :-------------------------: | :----------------------------: | :----------------------------: |
> > > > | Ours + C-Emb.            | $92.30_{\pm 0.05}$ |      $86.56_{\pm 0.08}$      |      $98.87_{\pm 0.00}$      |
> > > > | Ours                     | $89.34_{\pm 0.03}$ |      $81.56_{\pm 0.05}$      |      $98.78_{\pm 0.00}$      |
> > > >
> > > > While Ours performed forgetting for only the classes with the training samples based on the loss given in Sec. 3.2, Ours + C-Emb. performed forgetting for all the classes by incorporating the negative NT-Xent loss. We see that Ours + C-Emb. could outperform Ours in all the metrics, which proves the effectiveness of our recommendation.
> > > >
> > > > We would like to thank you for your question which led us to discover a promising extension of our approach, Ours + C-Emb. which was not discussed in our original paper! While the development of effective zero-shot prompt tuning is beyond the scope of this paper, we believe it is healthy that such a valuable discussion will arise in the community as a result of our paper, which is our pleasure. So we would like to add the above discussions in the final version.
> > > >
> > > > [a] Zhou et al., Learning to Prompt for Vision-Language Models. IJCV, Vol. 130, No. 9, pp. 2337–2348, 2022.
> > > >
> > > > [b] Menon and Vondrick, Visual Classification via Description form Large Language Models, in Proc. ICLR, 2023.
> > > >
> > > > [c] Cho et al., PromptStyler: Prompt-driven Style Generation for Source-free Domain Generalization, in Proc. ICCV, 2023.

---

> > > > ### Comment · Reviewer_LjLN · 2024-08-13
> > > >
> > > > __following on (c)__: I get the idea for word embeddings, or even when working with multiple modalities, where there are shared spaces across a modality, and specific features of each modality separately (like audio having tone specific features unused by text). However, within images, it is not so intuitive on how those spaces get disentangled between the shared and specific/unique based solely on the black-box output of classification layers, which represent orthogonal dimensions (due to the CE-loss). Specially if we think about how the black-box model does not provide information of the levels[^1] at which features are agreggated. This should have a direct effect on how knowledge relates within the model, and how it (or its paths) can be forgotten. It seems to require some knowledge about the output space characteristics of the black-box. How the knowledge is learned, like if the black-box is a mixture of experts, network in network, or other unknown settings, might also have an effect to how the shared or unique contexts have an effect.
> > > >
> > > > [^1]: here I mean field of view, scale of the pattern, or semantic level of the involved features.

---

> ### Comment · Reviewer_LjLN · 2024-08-13
>
> __following on (a) and (d)__: yes, that is what I meant. I also agree that it is a larger question than what the paper covers currently. I appreciate the results provided and the thorough explanation with the new results. I agree that the insights from the discussion are indeed of interest for further research!
>
> Overall, my questions have been answered. Thanks to the authors for their thorough responses. Some clarifications within the discussion would be needed for the final version, but I have no further questions. I think the questions from the other reviewers and the author responses to them are also satisfactory. Some of them highlight some of the limitations of the proposed setting and also should be included in a final version. The only comment from a reviewer that I also consider important to note, is how simple the datasets used are (Q4 from SKod), and agree with the observation. The results on ImageNet-30 by the authors are good step to solve that issue, although I would be cautious of results obtained during a rebuttal (not due to suspicion of malice, but just due to the fast implementation). However, I am satisfied with the experiments provided. Therefore, I would be in favour of accepting the paper.

---

> > ### Author Response · Authors · 2024-08-13
> > **Additional responses.**
> >
> > Thank you again for your feedback. We are pleased to hear that we were able to properly address your questions and that you are in favor of accepting our paper.
> >
> > **To (c)**: Thank you for drawing our attention to this interesting point. We will try to put more detailed discussions in the final version, but our preliminary idea in the case of pre-trained VLMs at present is as follows.
> >
> > Contrastive language-image pretraining learns to align image and text features in a common space using a massive dataset. Thus, the common and unique components of the text features are expected to be aligned with the image features as well, even if the modalities are different. Empirically, image-specific "levels" seem to be aligned with the text features, at least partially, but surprisingly very well. For example, it has been shown that the pre-trained CLIP can recognize an object in a local region of an image, suggesting that the text features are learned to absorb differences in scale [a]. CLIP guidance can produce an image that is faithful to the prompt by moving the image feature in the direction of the prompt embedding [b]. These results may support the earlier expectation and may also be a reason that effective forgetting is possible for pre-trained VLMs.
> >
> > **To (a) and (d)**: We will make sure to add the results on ImageNet-1K in the final version. Thank you.
> >
> > [a] Miyai et al., LoCoOp: Few-Shot Out-of-Distribution Detection via Prompt Learning, in Proc. NeurIPS, 2023.
> >
> > [b] Nichol et al., GLIDE: Towards Photorealistic Image Generation and Editing with Text-Guided Diffusion Models, in Proc. ICML, 2022.

---

### Author Rebuttal · Authors · 2024-08-06

We thank all the reviewers for their thoughtful review and constructive feedback. We are happy to see that the reviewers acknowledged the major contributions of this paper. Namely,

1. We proposed a novel task called Black-Box Forgetting, which aims to achieve selective class forgetting under the assumption that the parameters and gradients of the model are inaccessible.

2. Aiming at improving derivative-free prompt tuning for Black-Box Forgetting, we proposed a novel context parametrization method called Latent Context Sharing (LCS) that explicitly models shared and unique components over multiple latent contexts.

The main questions raised by the reviewers centered on 1) the difference between our task, i.e., Black-Box Forgetting, and typical machine learning, and 2) the requests for additional analyses to further support the validity of our method.
In the rebuttal, **we addressed all of the reviewers' questions with the support of thorough additional experimental results**. We will implement all of these responses in our final version. For the additional experimental results, please see the PDF attached to this thread.


### Black-Box Forgetting vs. Machine Unlearning.

These two are closely related but different. Machine Unlearning typically aims to remove the influence of specified training samples on the training model, whereas Black-Box Forgetting aims to prevent the recognition of specified classes. Furthermore, we in this paper address the black-box setting, which has not yet been well-traveled in machine unlearning. To explain this, we will add a brief review of Machine Unlearning in the Related Work section in the final version.


### Additional analyses to support the validity of our method

We additionally conducted through experiments and report the new results. Namely, evaluatin of error bars, comparisons with fine-tuned CLIPs, comparisons with ResNet and ViT trained from scratch, and evaluations on an additional dataset (ImageNet30), and so on. For more information, please see Tables A-H in the attached PDF and our responses to each reviewer's questions.

---

### Author Response · Authors · 2024-08-14

Dear ACs and Reviewers,

Reviewer-Author Discussion is almost over. Once again, we would like to thank ACs and Reviewers for devoting a great deal of time and effort to our paper. We have received many constructive comments from the reviewers through this discussion period, which significantly improves the final version of our paper. We believe that we were able to thoroughly address all of the questions posed by the reviewers with the best effort we could in this short rebuttal period.

Furthermore, as some of the discussions we have had here have, our paper has a potential to bring new perspectives to the machine learning community even beyond the scope of our paper, and serve as sources of inspiration for many researchers.

Many Thanks,

Authors

---

### Decision · Program_Chairs · 2024-09-25

**Decision:**

Accept (poster)

**Comment:**

This paper explores the selective forgetting problem for large-scale pre-trained models. In the paper, the authors introduce a novel method based on the derivative-free optimization of a learnable text prompt. Initially, this paper received mixed ratings. During the discussion period, the authors and the reviewers had in-depth discussions, and all the reviewers agreed that this paper is good enough to be presented at NeurIPS. The authors are required to include the discussions with all the reviewers in the final version.